# Retinal Vascular Endothelial Cell Dysfunction and Neuroretinal Degeneration in Diabetic Patients

**DOI:** 10.3390/jcm10030458

**Published:** 2021-01-25

**Authors:** Malgorzata Mrugacz, Anna Bryl, Katarzyna Zorena

**Affiliations:** 1Department of Ophthalmology and Eye Rehabilitation, Medical University of Bialystok, 15-089 Białystok, Poland; anna.bryl@umb.edu.pl; 2Department of Immunobiology and Environment Microbiology, Medical University of Gdańsk, 18-211 Gdańsk, Poland; katarzyna.zorena@gumed.edu.pl

**Keywords:** diabetes, diabetic retinopathy, retinal vessels, endothelium, dysfunction, retinal neurodegeneration

## Abstract

Diabetes mellitus (DM) has become a vital societal problem as epidemiological studies demonstrate the increasing incidence of type 1 and type 2 diabetes. Lesions observed in the retina in the course of diabetes, referred to as diabetic retinopathy (DR), are caused by vascular abnormalities and are ischemic in nature. Vascular lesions in diabetes pertain to small vessels (microangiopathy) and involve precapillary arterioles, capillaries and small veins. Pericyte loss, thickening of the basement membrane, and damage and proliferation of endothelial cells are observed. Endothelial cells (monolayer squamous epithelium) form the smooth internal vascular lining indispensable for normal blood flow. Breaking its continuity initiates blood coagulation at that site. The endothelium controls the process of exchange of chemical substances (nutritional, regulatory, waste products) between blood and the retina, and blood cell passing through the vascular wall. Endothelial cells produce biologically active substances involved in blood coagulation, regulating vascular wall tension and stimulating neoangiogenesis. On the other hand, recent studies have demonstrated that diabetic retinopathy may be not only a microvascular disease, but is a result of neuroretinal degeneration. Neuroretinal degeneration appears structurally, as neural apoptosis of amacrine and Muller cells, reactive gliosis, ganglion cell layer/inner plexiform (GCL) thickness, retinal thickness, and retinal nerve fiber layer thickness, and a reduction of the neuroretinal rim in minimum rim width (MRW) and functionally as an abnormal electroretinogram (ERG), dark adaptation, contrast sensitivity, color vision, and microperimetric test. The findings in early stages of diabetic retinopathy may precede microvascular changes of this disease. Furthermore, the article’s objective is to characterize the factors and mechanisms conducive to microvascular changes and neuroretinal apoptosis in diabetic retinopathy. Only when all the measures preventing vascular dysfunction are determined will the risk of complications in the course of diabetes be minimized.

## 1. Introduction

Diabetes mellitus (DM) has become a major societal problem. Epidemiological studies demonstrate the increasing incidence of type 1 and 2 diabetes mellitus (T1DM and T2DM, respectively) [1,2,3,4]. It is assumed that in the coming years the number of patients with diabetes will increase [1,5,6]. In 2012, the number of people with DM was 371 million, including 25.8 million in the USA. Globally, as many as 4.7 million patients died due to DM complications for the year 2012 [7].

Lesions observed in the retina with diabetes, referred to as diabetic retinopathy (DR), are caused by vascular abnormalities and are ischemic in nature. Due to its widespread prevalence, diabetic retinopathy is considered the major culprit of blindness in industrialized and middle-income countries [8]. There exists a clinical division of diabetic retinopathy into non-proliferative and proliferative retinopathy. The time of Lesion progression differs among patients and is determined by diabetes duration, glycemia, genetic predispositions and treatment methods. In the USA, it is calculated that among patients with type 2 diabetes T2DM 40.3% have DR, and 8.2% suffer from vision-threatening retinopathy [9]. In patients with type I diabetes T1DM 86% have retinopathy and 42% vision impairment due to DR [10]. Studies conducted on a group of 22,896 diabetic patients showed that 34.6% had DR, and the rising risk was associated with diabetes duration and improper blood glucose and blood pressure monitoring. Vision-threatening stages of DR involve proliferative DR and diabetic macular edema (DME). The incidence of proliferative DR and DME in the researched group totaled 6.96% and 6.81% respectively. Vision impairment related to DR is a serious global health problem. [11,12].

Although the clinical examination may confirm the retina’s normality, after a few years of DM some important histological and biochemical lesions usually appear, including adhesion of leukocytes, thickening of the basement membrane as well as loss of pericytes. Pericytes are the mural cells of blood microvessels, which have recently come into focus for modulating angiogenesis, regulating blood flow, and maintaining blood–retina barrier (BRB) integrity. Pericytes lying on the capillaries, and are surrounded by the basement membrane, they can prevent ischemia-reperfusion after thrombus clearance by constricting capillaries whereas their relaxation increases blood flow [13]. When the time of duration of diabetes increases, substantial vascular lesions are more likely to affect the retina. As the duration of diabetes increases, the probability of remarkable vascular alterations in the retinal tissue rises. Advancing dysfunctional process of endothelial cells plays a key role in the structure and pathophysiology of the retina, such as thickening of the basement membrane, loss of perivascular cells, damage to the BRB and neovascularization [14,15]. These alterations are accompanied by important biochemical processes, including formation of advanced glycation end-products, and activation of protein kinase C isoforms and the polyol and hexamine pathways. [15]. Subsequently, this contributes to oxidative stress, inflammation and vascular dysfunction. Vascular lesions in diabetes affect small vessels (microangiopathy) and involve precapillary arterioles, capillaries, and small veins. The decreasing number of pericytes, thickening of the basement membrane, and proliferation of endothelial cells are observed [16,17].

## 2. The Retina—Hyperglycaemia and Inflammation in the Course of Diabetes

Many years of research have shown that hyperglycaemia plays a central role in the induction of diabetic retinopathy [18,19,20]. Studies conducted in non-obese diabetic mice (NOD mice) have demonstrated that the first changes on the fundus of the eye blood–retinal barrier breakdown are observed already in the first week of exposure to high glucose levels [21]. In the environment of high glucose concentration, hyperglycaemia causes cell dysfunction, retinal neurovascular impairment, structural defects and functional disorders which lead to further damage of the retinal cells [13,18,19]. The next active stage of pathological changes is associated with the inflammatory process [22,23,24]. The key factors of the inflammatory process in diabetes, first local, then systemic and chronic, are chemokines, growth factors and cytokines [25,26]. In the course of inflammation, acute-phase proteins, including C-reactive protein (CRP) are produced in response to cytokine stimulation. Under physiological conditions, the level of CRP synthesis is low, however, the production increases in inflammation and it is observed in many inflammatory diseases, including diabetes [27,28,29]. Previous studies have demonstrated the elevated blood levels of CRP in patients with T1DM and T2DM suffering from diabetic complications, such as diabetic retinopathy (DR) [18,24,25,26,27]. In patients with T1DM and retinopathy, a five-fold higher level of CRP protein was detected in the blood serum compared to the group of patients with T1DM and without diabetic retinopathy [27]. The authors conclude that the persistently elevated levels of proinflammatory cytokines and CRP in chronic diabetes result from an ongoing inflammatory process in diabetes [25,26,27]. CRP is a clinically recognized marker of inflammation, however, other proteins are also proposed as useful markers of diabetic retinopathy. For example, in patients with T2DM, interleukin 34 (IL-34) has been shown to be an additional inflammatory marker in predicting the risk of chronic diabetic complications. The IL-34 parameter was found to have better discriminate values for the risk of chronic diabetic complications than the CRP protein. Based on the order of the discriminate power, defined as the area under the curve, it was found that the AUC_ROC_ area was greater for IL-34 (AUC = 89.88%) than for CRP protein (AUC = 83.96%) [26].

In other studies, the authors attempted to demonstrate whether selected inflammatory proteins may be associated with microvascular complications in adult T1DM patients [29]. In a group of 100 subjects with T1DM, the following parameters were determined: epidermal growth factor (EGF), metalloproteinase 2 (MMP-2), growth/differentiation factor 15 (GDF-15) and interleukin 29 (IL-29). Screening was performed for microvascular complications, such as autonomic and peripheral neuropathy, diabetic nephropathy, and diabetic retinopathy. The results of multivariate logistic regression showed that an increase in the EGF concentration was a statistically significant predictor of microangiopathy (*p* < 0.0001). Moreover, higher levels of GDF-15 have been associated with diabetic nephropathy, peripheral neuropathy and proliferative retinopathy rather than with non-proliferative retinopathy in patients with T1DM [30].

On the other hand, in children and adolescents suffering from T1DM and diabetic retinopathy, a higher level of IL-6 was demonstrated in the blood serum [31]. The authors showed a significant gradual increase of the IL-6 serum level. This was demonstrated by comparing the values in healthy children, children with T1DM without retinal changes the organ of sight and a group of children with the symptoms of non-proliferative diabetic retinopathy [31]. Higher serum levels of IL-6 were also shown in patients with T2DM and proliferative diabetic retinopathy rather than in the group of patients with T2DM without complications [32,33]. The presence of chronic inflammatory environment in the course of diabetes increases the expression of inflammatory factors, also in the aqueous humour of the eye [22,34]. The eight following factors have been recently found in the aqueous humour of DR patients: interleukin IL-6, IL-8, IL-10, vascular endothelial growth factor (VEGF), transforming growth factor-β (TGF- β), vascular cell adhesion molecule-1 (VCAM-1), intercellular adhesion molecule-1 (ICAM-1), and monocyte chemoattractant protein-1 (MCP-1) [13]. The study showed that TGF-β, ICAM-1, IL-10, VEGF, and VCAM-1 may play a role in the progression of diabetic retinopathy. The authors suggest that the cytokines could be potentially used as biomarkers for predicting the progression of diabetic retinopathy and help to choose a therapeutic option and/or monitor response to treatment [22,23].

Tumour necrosis factor alpha (TNFα) exerts a significantly strong effect on the development and progression of diabetic retinopathy [35,36,37,38]. Tumour necrosis factor alpha (TNFα), also known as TNF, cachectin, or differentiation inducing factor (DIF), is a pleiotropic proinflammatory cytokine, as well as one of 22 proteins belonging to TNFα superfamily, regulating cell growth and differentiation. Apart from its participation in inflammatory processes, it plays an essential role in angiogenesis. TNFα may have an inhibiting or stimulating effect on the formation of new vessels. The resulting effect of TNFα is most probably dependent on cell exposure time and its local concentration. Using a non-obese diabetic mice (NOD mice) model, it was demonstrated that the administration of TNFα into the vitreous body of the eye causes endothelial ischemia and retinal necrosis. This finding proves TNFα role in the pathogenesis of diabetic complications [37]. On the other hand, clinical studies in the group of type 1 diabetic children showed that TNFα turned out to be the paramount tested factor [35]. Studies conducted have demonstrated a significantly higher level of serum TNFα in 76% of children with T1DM and with NPDR compared to the group of children without DR (35%), as well as compared to healthy control group, in which no serum TNFα was detected. Moreover, findings indicated that from within the proinflammatory factors tested, serum TNFα level may be an independent indicator in the prediction of NPDR development in children [35]. Other authors have also detected a high blood serum TNF level in adult patients suffering from T1DM. Authors, using a multifactorial analysis of logistic regression, have proven that TNFα was an autonomous determinant of the PDR inflammatory state marker [36]. The last study tested the level of cytokines in the vitreous body and their correlation with the inflammatory cell density in the fibrovascular membranes (FVM) in patients with proliferative diabetic retinopathy (PDR) in order to assess intraocular inflammatory states in relation to the disease activity [38]. The authors’ statistical analysis demonstrated that PDR-affected patients had significantly higher levels of monocyte chemoattractant protein-1 (MCP-1) (*p* = 0.003), VEGF (*p* = 0.009) and interleukin 8 (IL-8) (*p* = 0.02) in the vitreous body compared to patients with inactive PDR. Moreover, statistical methods confirmed a significantly greater number of T lymphocytes (CD3+, CD4+ and CD8+) in PDR patients compared to PDR ones. The authors suggest that a relationship between the level of cytokines (MCP-1 and IL-8) in the vitreous body and the inflammatory cell density in FVM, and differences in cytokine levels in the vitreous body between PDR and without PDR groups of patients indicate the importance of local intraocular inflammation in PDR patients [39].

## 3. The Ischemia, Hypoxia and Neoangiogenesis in the Development and Progression of Diabetic Retinopathy

Studies have demonstrated that in the healthy individual’s eye vascular endothelial cells are mitotically inactive and the vascular growth factors stay in balance with anti-angiogenic factors [40,41,42]. Under physiological conditions, eye maintains the perfect balance of angiogenic modulators [43,44,45]. However, under some circumstances, such as hypoxia or inflammation, the equilibrium may be shifted towards angiogenesis—a phenomenon known as angiogenic switch [41,42]. Ocular angiogenesis is a complex multi-stage process resulting in the formation of new blood vessels from an existing so-called “vascular tree”. This usually leads to a considerable loss of eyesight in both T1DM and T2DM patients [46,47]. Possible mechanism of retinal vascular endothelial cell dysfunction and neuroretinal degeneration in diabetic patients is presented in Figure 1 [42,43].

In the course of studies, it has been demonstrated that hyperglycaemia leads to hypoxia and inflammation. The hyperglycaemia elevates hypoxia induced factor 1 (HIF-1) and insulin-like growth factor 1 (IGF-1), both in the serum and the vitreous body of diabetic patients [48,49,50]. The overexpression of HIF-1, IGF-1, and other factors not only activates Müller cells to form a chronic inflammatory milieu, but also induces the overexpression and accumulation of VEGF and fibroblast growth factor (FGF), thus initiating retinal fibrosis and pathological neovascularization [49,50].

There are solid experimental and clinical proofs, confirmed by our research team, of the role of vascular endothelial growth factor (VEGF) in the retinal angiogenesis. The retinal hypoxia leads to a marked elevation of VEGF in several types of cells, including pericytes, Müller cells, astrocytes, endothelial cells of the retina and pigment epithelium of the retina [41,42,51,52]. VEGF is the strongest factor stimulating physiological and pathological angiogenesis. It is a glycated homodimer with the molecular mass of 46–48 kDa [42]. VEGF is synthesized by endothelial cells, macrophages, lymphocytes (CD4), plasmatic cells, myocytes, megakaryocytes and neoplastic cells. VEGF stimulates endothelial cell proliferation and migration, and enhances blood vessel permeability [51,52,53]. Furthermore, it induces the production of tissue collagenase and increases the macrophagous and monocytic potential for chemotaxis [42]. VEGF expression precedes retinal neovascularization in the retinas and the optic nerves of humans with diabetes. Its localization to glial cells of the inner retina and the anterior optic nerve suggests a relationship to neovascularization in these sites. That VEGF immunopositivity may occur when there is no morphological evidence of retinal nonperfusion and little likelihood of retinal neovascularization suggests the possibility that ischemia may not be the only stimulus for VEGF expression [54]. In addition, VEGF immunoreactivity is correlated with increased vascular permeability, as indicated by human serum albumin (HSA) immunostaining, and appears to be increased in diabetic subjects before the onset of retinopathy [55].

Higher levels of VEGF were detected in children and adolescents with T1DM and PDR, but also in patients, in whom an ophthalmologic examination did not reveal any ocular lesions [51,52,53]. Obtained results suggest that VEGF may take part in the development of vascular changes within the visual organ in children and adolescents as early as during first years of the disease, when standard diagnostic methods are still insufficient to detect any lesions of diabetic retinopathy nature. Additionally, other studies showed that VEGF level was higher in patients representing all three complications, i.e., hypertension, retinopathy and nephropathy, compared to diabetic patients without hypertension but with retinopathy and nephropathy. Moreover, no statistically significant differences were demonstrated in the level of VEGF between the group of patients with T1DM, retinopathy and nephropathy but without hypertension compared to a healthy control [53]. These data indicate that until no vascular complications are present in T1DM patients, VEGF level is not significantly higher in comparison with the healthy control group [52]. Moreover, an augmented level of VEGF was detected in the aqueous humour of the eyes of adult patients suffering from the proliferative form of diabetic retinopathy [22,56,57]. Moreover, it was found that an augmented level of VEGF in the aqueous humour of the eyeballs of patients suffering from the proliferative form of diabetic retinopathy also correlates with an increased level of serum VEGF [56,57]. As inferred from findings mentioned above, VEGF is involved in the development of diabetic retinopathy both in young diabetes patients as well as in adult T1DM and T2DM patients.

Recent studies found out that chronic exposure of the retina to hyperglycemia leads to accumulation of advanced glycation end products (AGEs) that play an important role in retinopathy. The receptor for AGEs (RAGE) is expressed in various retinal cells and is upregulated in the retinas of diabetic patients. AGEs can exert their deleterious effects by acting directly to induce aberrant crosslinking of extracellular matrix proteins, to increase vascular stiffness, altering vascular structure and function. Moreover, AGEs binding to the receptor for AGEs (RAGE) evokes intensive intracellular signaling cascades that leading to endothelial dysfunction, elaboration of key proinflammatory cytokines and proangiogenic factors, such as interleukin 1 (IL1), tumor necrosis factor α (TNF-α), and vascular endothelial growth factor (VEGF), adhesive molecules, and the activation of a nuclear transcription factor NFκB, mediating pericyte apoptosis, vascular inflammation, and angiogenesis, as well as the breakdown of the inner blood–retinal barrier (BRB). The end result of all these events is damage to the neural and vascular components of the retina Figure 1 (Panel C) [18,42,43]. AGE-RAGE axis plays a crucial role in the sustained inflammation, neurodegeneration, and retinal microvascular dysfunction occurring during diabetic retinopathy [58,59].

## 4. The Neurovascular Unit of the Retina

The retinal neurovascular unit includes the physical and biochemical relationship among neurons, glia, and blood vessels and the close interdependency of these tissues in the retina and the central nervous system. The neural unit (ganglion cells and glial cells) and the vascular unit (endothelial cells and pericyte) constitute the retinal neurovascular unit, together with retinal pigment epithelial (RPE) cells, they are the main body of BRB. The BRB is divided into two parts: the inner blood–retina barrier (iBRB) consists of retinal endothelial cells that are covered by astrocytes, pericytes, and Müller cells end-foot and is essential for maintaining the microenvironment of the inner layers of the retina. The outer blood–retina barrier (oBRB) is composed of tight junctions formed by neighboring RPE cells and serves as a filter to regulate solutes and nutrients from the blood. Hyperglycemia causes cells dysfunction, when impairment occurs, structural defects and functional disorder arise in the neurovascular unit, which in turn lead to the further impairment of cells [13,60,61,62].

### 4.1. The Neural Unit

The neural unit includes retinal ganglion cells (RGCs), glial cells, and other neural cell types.

#### 4.1.1. RGCs

The Integrity of RGCs provides the normal function of the retina. However, abnormal environment such as hyperglycemia alters the structure or function of RGCs and this kind of RGCs impairment is progressive in the development of subsequent DR.

#### 4.1.2. Glial Cells

There are three major types of glial cells in the retina and they are involved in maintaining retinal homeostasis: Müller cells, astrocytes, and microglia.

Müller cells are the leading type of glial element, representing 90% of the retinal glia. The inner limiting membrane is made up by the bases of Müller cells and connects to the base of the vitreous humor. The circular junctions between photoreceptors and Müller glial cells form the outer limiting membrane. Müller cells manage vascular responses to fulfill the metabolic demand of neurons, interchange metabolites, recycle neurotransmitters, and aid establishing the extracellular chemical environments [63]. It has been proven that the population of these cells shows traits of stem cells and is able to develop in different cells of the retina. The accelerated apoptosis of ganglion cells is accompanied by lesions within Müller glial cells.

Hyperglycemia and inflammation induce activation of microglial cells that are found in the inner part of the retina and migrate later to the subretinal space and release cytokines, thus causing death of nerve cells. The activated microglial cells adhere to the vessels and seem to play a crucial role in vascular wall damage [64,65]. Similar results were obtained by Schellini et al., who investigated the morphology of Müller cells in the retina of healthy rats, with treated and untreated diabetes, one and 12 months following the induction of diabetes. In treated rats the lesions were less distinct [66,67,68]. It was also found that Müller cells in diabetic patients showed increased expression of glial acidic fibrillary protein (GFAP) [69]. Since Müller cells produce factors able to modulate blood flow, vascular permeability and cell survival, they are thought to have a major effect on endothelial function.

The connection between Müller cells and the vascular unit is affected during DR and they produce large amounts of factors, while some show protective effect, more are making it worse such as vascular endothelial growth factor(VEGF). VEGF is upregulated in the early stages of DR and presents a neuroprotective effect for the maintenance and survival of neurons, VEGF derived from Müller cells in diabetes can also increase blood circulation for proper metabolic activities [54,55].

Named based on their stellate shape, astrocytes are linked to the retinal blood vessels and neurons and play a significant role in the BRB integrity. In contrast to Müller cells, astrocytes are considered almost exclusively located in the innermost retinal layers. The distribution of astrocytes is related to the presence and position of retinal blood vessels, therefore, astrocytes are more likely to locate in the vascular regions of the retina, while avascular zones contain nearly no astrocytes [70]. Under the diabetic environment, characterized by dysmetabolic and hypoxic, astrocytes are activated and produce a variety of pro-inflammatory cytokines as well as chemokines, together with T-cells, monocytes/macrophages, and microglia, amplifying the inflammatory response [71].

As the resident inflammatory cells of the retina, in addition to immunologic activity, microglia also participate in neural network development and maintenance. In non-proliferative DR(NPDR), the increased perivascular microglia cells settled into the retinal plexiform layers and exhibit mildly hypertrophic. In PDR, microglial cells gathered in ischemic areas and new dilated blood vessels heavily. The contact between microglia and RGCs are also found in epiretinal membranes. These findings suggest that activated microglia are involved in all stages of DR and may even accelerate DR progression [72].

### 4.2. The Vascular Unit

Micro-vasculopathy is a typical characteristic of DM, which can eventually lead to vision loss. As the basic unit of retinal vascular structure and function, the stability of pericytes and endothelial cells is necessary for maintaining vascular integrity.

#### 4.2.1. Endothelial Cells (EC)

Endothelial cells (EC) (monolayer squamous epithelium) form the smooth internal vascular lining indispensable for normal blood flow. They play a major role in cardiovascular homeostasis maintenance, mainly through the regulation of arterial blood pressure. Apart from ensuring the barrier between the vessels and lumen, endothelial cells secrete numerous mediators regulating platelet aggregation, fibrinolysis, coagulation, and the tension of the vessels. Endothelial dysfunction observed in diabetic patients contributes to the development of lesions in large and small vessels. The major factors include hyperglycemia, insulin resistance, hyperinsulinemia and dyslipidemia. Endothelial cells secrete a few mediators that can alternatively mediate in vascular stenosis [73,74,75]. The complex tight junctions of ECs are one of the significant parts of the iBRB, which protects the retina from injuries by circulating substances or metabolites. Under the development of DR, increased capillary occlusion results in retinal ischemia and hypoxia, triggering an increase in the levels of VEGF, which then promotes pathological neovascularization. The receptors on ECs are activated by VEGF, and this kind of activation promotes ECs survival, proliferation, permeability, and taxis toward the VEGF sources, the essential steps of blood vessels production and maintenance. However, in many DR patients, VEGF is excessively produced in the retina resulting in the over-proliferation of ECs, new blood vessel formation, thus contributing to microaneurysms, fluid leakage, and tissue damage [76].

Endothelial progenitor cells (EPC) are immature cells able to differentiate into mature endothelial cells. Vascular damage and tissue ischemia mobilize the EPC by releasing growth factors and cytokines. The EPC stimulate compensatory angiogenesis chiefly at the sites of ischemia in the peripheral circulation. Furthermore, the EPC make up a circulating pool of cells capable of forming a “cellular patch” at sites of endothelial damage, simultaneously contributing directly to homeostasis and repair of the endothelial layer. Tissue ischemia is regarded as the most powerful stimulus of EPC mobilization by means of activation of the systems that detect hypoxia, such as hypoxia-induced factor (HIF) -1α. Both patients with type 1 and 2 diabetes have fewer circulating EPC than healthy individuals. EPC in diabetic patients show the impairment of the function, such as decreased proliferation, adhesion, migration and a potential to be incorporated into vascular structures [13,74,75].

#### 4.2.2. Pericytes

Pericytes are the mural cells of blood microvessels, which have recently come into focus for modulating angiogenesis and maintaining BRB integrity [77]. Pericytes lying on the capillaries and are separated from epithelial cells by the basement membrane. They can prevent ischemia-reperfusion after thrombus clearance by constricting capillaries whereas their relaxation increases blood flow [78].

Loss of pericytes, accompanied by ECs degeneration and the basement membrane thickening, may leads to the formation of acellular capillaries. These abnormal capillaries are inclined to generate microaneurysm and hemorrhage, and are related to vascular occlusion, or induced by thrombosis secondary to residual ECs impairment, resulting in ischemic-hypoxic retinal damage [79].

## 5. Genetic Background and Biochemical Transformations

It is not fully clear what mechanisms determine the vulnerability of diabetic patients to retinopathy. There is a lot of evidence suggesting that this ailment can be related to gene polymorphisms of factors taking part in angiogenesis, such as the VEGF, SDH, AR, SDF-1, and TIMP-3 genes. It is thought that vascular changes are initiated with a major role of aldose reductase (AR; ALDR1) and sorbitol dehydrogenase (SDH; SORD), the components of a two-stage metabolic pathway transforming glucose to fructose. Aldose reductase (AR) is an enzyme converting glucose to sorbitol via reduction, with the use of NADPH. The presence of AR can be observed in the liver, Schwann’s cells, peripheral nerves, placenta, seminal vesicles, erythrocytes, lens and retina of the eye. The elevated level of aldose reductase is correlated with diabetes complications. It has also been proved that AR takes part in retinopathy development. Eight alleles of the polymorphic gene AR have been identified. Among of these genes, the *Z-2* allele may increase the risk of the development of retinopathy in Caucasian patients with type 2 diabetes, which was confirmed by Petrowic et al. who compared a group of diabetics with developed retinopathy with patients suffering from diabetes of 10 years’ duration without retinopathy [80,81,82,83].

Damage to endothelial cells and the loss of pericytes trigger the processes that narrow the vascular lumen, and thus reduce the blood flow. This in turn precipitates retinal ischemia and hypoxia. The hypoxic retina produces HIF, which is a transcription factor. HIF stimulates, among others, the release of vascular endothelial growth factor (VEGF) to vascularize the areas affected by ischemia. Next, the circulating VEGF binds to VEGF receptors on the vascular endothelial cells of the retina, triggering the pathway of tyrosine kinase that leads to angiogenesis. This results in the appearance of new vessels on the retina and optic disc (proliferative retinopathy) [84]. The increased level of VEGF leads to an increase in vascular permeability and to the formation of retinal edema. Proliferative retinopathy development can involve local endothelial cells as well as bone marrow-derived EPCs. Thus, vascular ischemia may be accompanied by pathological neovascularization in diabetic patients [54,55,85,86].

Additionally, higher levels of IL-6, IL-7, and IL-8, that cause increasing adhesiveness of leucocytes to vascular endothelium, have been observed in the vitreous and aqueous humour. It is considered that chronic inflammation of the vascular walls also encourages their occlusion and intensifies retinal ischemia [87,88].

## 6. The Role of Nitric Oxide

Nitric oxide (NO) is probably the major mediator of endothelium-derived particles, and its reduced bioactivity is the first and foremost endothelial dysfunction marker.

In diabetes, the biochemical pathways related to hyperglycemia may enhance the production of free radicals through a reduction in the amount of biologically active NO. The activation of protein C kinase, the depletion of nicotinamide adenine dinucleotide phosphate and the formation of advanced glycation end products in diabetes cause NO reduction. In hyperglycemic conditions, the enhancement of glucose metabolism in endothelial cells is assisted by increased production of free oxygen radicals. NO produced in small amounts in vascular endothelium plays a role of a signaling molecule which regulates the tension of the wall of the vessels. NO exhibits destructive properties towards other cells when generated in large amounts by inducible NO synthase in macrophages. The production of free oxygen and nitrogen radicals is under control over the antioxidant system. In the case when this system does not work, an imbalance occurs between pro- and anti-oxidant processes, which is referred to as antioxidative stress [89]. In addition, the changes within mtDNA caused by reactive oxygen species (ROS) leads to mutagenesis in the mitochondria and surrounding structures, which are in charge of the production of endogenous reactive oxygen. This is a cause of the decreased production of energy below the level required for tissue functioning, which results in tissue dysfunction or even premature death as antioxidant defence mechanisms become triggered with the aim to minimize the harmful effects exerted by reactive oxygen species [90].

## 7. Endothelin (ET)

The ET system embraces three vasoactive and neural peptides (ET-1, ET-2, and ET-3), two G-protein-coupled receptors (ETA and ETB), and two ET converting enzymes (ECE-1 and ECE-2). Peptides are coded by three different genes. ETs are considered to be paracrine hormones exerting an effect locally at the site of synthesis.

ET-1 is the major cardiovascular isoform of the ET system produced mainly in endothelium, even though it can be generated in vascular smooth muscle cells (VSMC) of the arterial wall, macrophages, leukocytes, cardiomyocytes, and fibroblasts too [91]. Murata et al. found no signs of ET-2 gene expression in the retina [92]. The function of ET-3 remains nebulous. ETs exert an effect on specific receptors, found in eye tissue and vessels of the retina. It has been demonstrated that ET-1 is a potent agent narrowing vessels with mitogenic, prooxidative and proinflammatory properties influencing the regulation of vascular function, including the development of DR. Overproduction and increased functional effects of ET-1 are reported to be strongly altered in diabetic conditions. The ETA and ETB receptors, coupled with two distinct G-proteins, mediate the action of ET-1 on the vascular tone. The two receptors in combination with two separate G proteins affect ET-1 on vascular tension. In patients with diabetes, signaling mechanisms change between ET-1 and receptors. ET-1 is synthesized by the transforming growth factor β and is one of the most powerful factors narrowing the vessels and its action has a double effect: it affects the EC and pericyte receptors but also it has a mitogenic effect on smooth muscle cells.

Therefore, a fall in ET-1 production or its action influences the integrity and stability of VSMCs partly responsible for the characteristic loss of pericytes [93].

Hyperglycemia induces oxidative stress and upregulation of ET-1, but, simultaneously, it activates numerous signaling pathways in nucleus of ECs and VSMCs, leading to the gene expression of lots of peptides and factors. In 2013, using both vivo and vitro systems, Feng et al. proved that miRNA-1 is downregulated in ECs under exposure to high-glucose blood levels. This contributes to the upregulation of ET-1, which has widespread downstream effects on genes of extracellular matrix: fibronectin and laminin, and evokes the thickening of the capillary basement membrane in DR [94].

The prevention of diabetic vascular complications seems contingent on early detection and treatment of endothelial dysfunction. Reduced availability of nitric oxide (NO) is a major characteristic of endothelial dysfunction at the initial stage of type 2 diabetes. Measurement of flow mediated dilatation (FMD) of the brachial artery after ischemia is to be a non-invasive method used to assess endothelial production and NO release. Impairment of reactive hyperemia resulting from microvascular dysfunction in diabetes can lead to inadequate increase in shear stress stimulating the release of endothelial NO [87].

A sphygmomanometer is situated over the arm to stimulate flow in the brachial artery. The sphygmomanometer is inflated until the systolic pressure exceeds 50 mm Hg, in this way arresting blood flow and causing ischemia. In consequence, the vessels become dilated in distal resistance arteries up to the sites where the flow is blocked. When the sphygmomanometer is emptied, reactive hyperemia appears in the brachial artery. The percentage difference between the diameter measured after reactive hyperemia and the basic diameter is referred to as flow-mediated dilatation (FMD). This method allows monitoring the effects of the treatment of endothelial dysfunction. Research has demonstrated that angiotensin converting enzyme (ACE) inhibitors, angiotensin 1 (AT1) receptor blockers, the newest generation beta-adrenolytics, such as carvediol and nebivolol, statins, estrogens, exercise, and diet increase FMD [88].

Unfortunately, FMD reflects endothelial functions in comparatively large vessels. In diabetic retinopathy, the mechanisms of reactive hyperemia in brachial artery microcirculation differ in the microcirculatory retinal system. In this case, the neurohormonal mechanism’s role is vital [89].

The available literature offers a description of the relationship between vascular endothelial function and diabetic microangiopathy (nephropathy, retinopathy, and neuropathy) in patients with type 2 diabetes as well as between endothelial function and macroangiopathy evaluated by the intima-media thickness complex (IMT). The endothelial function was assessed by means of FMD in the brachial artery. A considerable decrease was observed in FMD in patients with proliferative diabetic retinopathy as compared to those without. The FMD showed substantial negative correlations with systolic blood pressure and diabetes duration, although no correlations were found between FMD and IMT [90].

As mentioned previously, the vascular endothelial cells play a fundamental role in human homeostasis through the regulation of arterial blood pressure, and distribution of nutrients and hormones. It also provides a surface which modulates coagulation, fibrinolysis and inflammation. Endothelium dysfunction observed in diabetic patients contributes to the development of lesions in large and small vessels. The major factors include hyperglycemia, insulin resistance, hyperinsulinemia and dyslipidemia. Considering the impact of insulin analogues (empagliflozin, canagliflozin), studies are limited. Among the drugs that reduce lipid level, statins improve endothelial function (EF) in diabetic patients, whereas short-term research data indicate that ezetimibe and fenofibrate improves EF, however, further investigations are needed in diabetic patients. The action of acetylsalicylic acid on EF is dose-dependent, i.e., EF is improved by lower but not higher doses.

## 8. A Potential Link between Retinal Neurodegeneration and Microvascular Dysfunction

While vascular retinal lesions in the diabetes process have been explored well, lesions in nerve cells of the retina are still under clinical research. Until recently, the majority of researchers were focusing on vascular retinal lesions with the presumption that they were responsible for the change in neuronal function conditioned the neurons function change. In recent times, findings have been suggesting that the changes of the neuronal function and their viability contribute to the formation of DR pathogenic mechanisms starting soon after diabetes appearance [63,95,96,97]. Diabetes induces neural apoptosis of ganglion, amacrine, and Muller cells, activation of microglia caused by chronic glutamate toxicity, inflammatory glial activation, ganglion cell layer/inner plexiform (GCL) thickness, retinal thickness, and retinal nerve fiber layer thickness [97,98,99,100,101,102,103]. Piona et al. [104] reported that the reduction of the neuroretinal rim in minimum rim width (MRW) which is the shortest distance between Bruch’s membrane opening (BMO) and internal limiting membrane (ILM), may be a potential early marker of retinal degeneration detectable in T1DM-children. A relationship between MRW and mean HbA1c suggests that the control of glucose metabolism may affect an early neurodegeneration of the retina, starting from childhood. These retinal findings cause functional changes, which precede clinical DR and may occur prior to the diagnosis of DM. Functional changes include deficits in pattern electroretinogram (pattern ERG), increased implicit timed in the multifocal ERG (mf ERG), abnormal dark adaptation, and abnormal contrast sensitivity [105,106,107,108,109,110,111,112]. Recent years have delivered numerous proofs supporting the hypothesis that DR is a neurodegenerative complication [113,114,115]. The latest eight-year observations have revealed that the reduction in the thickness of the internal retinal layer takes place in T1DM patients along with the course of the disease but still before the DR manifestation. This confirms the hypothesis of retinal neurodegeneration [116].

The mediators engaged in the link between neuroretinal degeneration and microvascular changes have not been recognized yet. It is known that VEGF is significant in the early stages of DR. On the other hand, recently, a growing interest has been concentrated on the role of some other important molecules, such as ephrins, semaphorins, and netrin, which are released early by damaged neurons and may stimulate the development of DR [117,118,119]. The before mentioned molecules are highly expressed in the retina and vitreous in patients with advanced DR. Interestingly, according to the recent studies, early neurodegeneration in the retina may appear in diabetic patients who have no clinically proven DR (NRD) or mild non-proliferative DR (MDR). The evaluation of the relationship between microperimetric (MP) sensitivity and retinal thickness measurement have found the structural and functional abnormalities. There were significant total retinal thickness (TRT), inner retinal thickness (IRT) and MP sensitivity reductions observed in individuals having no diabetes or mild retinopathy [109,120]. In addition, patients with type 2 diabetes with no diabetic retinopathy but good metabolic parameters exhibited neurodegeneration affecting neurons in the macula and axons in the optic nerve. Systemic vascular changes led to further damage of the axons and indicated a potential role of subclinical ischemia in retinal neurodegeneration appearance [121].

It is thought that the most important mechanisms playing a role in the retinal neurons lesion process accompanying diabetic retinopathy are: extracellular glutamate accumulation, oxidative stress and a reduction of retinal neuroprotective factors synthesis. It is worthy of consideration that proteomic profiles analysis belonging to neurotransmitters of early DR revealed mediators of neurodegenerative brain diseases, such as Alzheimer’s and Parkinson’s disease [100].

Glutamate belongs to the main neurotransmitter of the retina. Its concentration jest elevated in extracellular space in the aqueous humour as well as the vitreous humor. Glutamate excess leads to increased activation of ionotropic glutamate receptors, which pass calcium ions to intracellular space of postsynaptic neurons uncontrollably and thereby cause cells death. In the case of diabetes, glutamate accumulation in extracellular space is primarily provoked by its diminished absorption via glial cells, inhibition of the activity of glutamine synthetase transforming glutamate to glutamine, and the depletion of retinal cells’ ability to oxidize glutamate to α-ketoglutarate

It has been demonstrated that oxidative stress is likely to damage retinal microvascular and neuronal cells (especially RGCs) and. The impairment of L-glutamate/L-aspartate transporter (GLAST) favoring excitotoxicity seems to be a potential mechanism. Glutamate toxicity leads to glutathione depletion having an impact on oxidative stress. In the light of these findings, oxidative stress appears to be an underlying mechanism connecting neurodegeneration with early microvascular abnormalities [103].

## 9. Neuroprotective Factors

Diabetic patients are characterized by the lower production of neuroprotective factors: pigment epithelial-derived factor (PEDF), somatostatin (SST) and interstitial retinol-binding protein (IRBP) in the retina in comparison to nondiabetic patients. Downregulation of these factors compromises neuroprotection against neurotoxic factors engaged in neuroretinal degeneration. PEDF is primarily synthesized by RPE and impedes angiogenesis and neurodegeneration by means of oxidative stress reduction in the retina or the expression of glutamine synthase increase, hence providing protection against glutamate excitotoxicity [122,123,124]. It has been recently presented that PEDF peptide eye drops diminish ganglion cell death, microglial activation and vascular leakage in rats with diabetes and therefore seem to be prospective treatment for early DR [125]. Furthermore, PEDF plays a significant role in homeostasis of the retina due to its antiangiogenic and neuroprotective activities and precludes oxidative stress and glutamate excitotoxicity. Thus, PEDF downregulation occurring in the diabetic retina appears essential for favoring neurodegeneration, and may mitigate initial microvascular abnormalities. Similarly to PEDF, SST exhibits antiangiogenic and neuroprotective properties and is chiefly synthesized by RPE. DME and PDR represent lower level of SST production, which is related to a crucial decrease in its intravitreal levels. Moreover, the downregulation of SST production in the human retina happens in the initial stages of DR and is attributable to retinal neurodegeneration [125].

Likewise, VEGF has neuroprotective properties. Except for downregulation of natural neuroprotective factors generated in the retina, an upregulation of neurotrophic and survival factors such as VEGF and erythropoietin (Epo) is also observed in the retina. Manifestly, this overexpression can be already detected in the early stages of DR without a predominance of ischemia [119].

Interlinking toxicity status involving glutamate and VEGF-induced blood–retina barrier breakdown is very important intriguing strands linking the phenomena of neuroretinal degeneration and vascular complications. Hyperglycemia is known to evoke extracellular glutamate increase and subsequently, NMDA receptor overactivity, which contributes to the accelerated neuron death. Perhaps, the anoxic retina aiming to spare neurons begins to produce the Hypoxia-inducible factor, which stimulates VEGF. VEGF as a neuroprotector is supposed to protect nerve cells by triggering angiogenesis, but it damages the blood–retina barrier while increasing vascular permeability. The mechanism is also grounded on glial cells. Their increased activity in the early stages of diabetes might contribute to neurons damage. In addition, the decrease in neuroprotective factors (PEDF, SST) might aid the disruption of the blood–retina barrier directly or by enhancing VEGF expression [119,123].

## 10. Conclusions

Summing up, apart from the conventional cardiovascular risk factors, endothelium dysfunction is essential in the early pathophysiological processes of vascular complications. Not only does it play a role of a passive barrier for blood vessels, but it also performs a number of important physiological functions, mediating the release of vasoactive factors regulating vascular wall tension, cell growth, homeostasis, and inflammatory conditions. In general, the term dysfunction denotes the inability of the endothelial cells to maintain normal homeostasis of vascularity and can be a crucial marker of altered vascular reactivity in diabetes. Only when all the measures preventing vascular endothelial dysfunction are determined will the risk of complications in the course of diabetes be minimized. It should be also remembered that high arterial blood pressure, elevated serum lipids and tobacco smoking promote retinal abnormalities [126,127]. Thus, not only does the adequate treatment matter in the prevention of diabetic retinopathy, but also patient’s lifestyle and diet. The available research data highlight the necessity to optimize the ABC criteria: A—serum hemoglobin A1C (HbA1C), B—blood pressure, and C—serum cholesterol [119]. The American Diabetes Association (ADA) recommends a reduction in HbA1C to <7%, maintenance of arterial blood pressure at <130/80 mm Hg and serum LDL <100 mg/dl (<70 mg/dl in patients with diagnosed cardiovascular disease). Such action will allow for a reduced risk of microvascular and cardiovascular complications also involving endothelial dysfunction [116]. In individuals who had increased their physical activity and changed eating habits a decrease was observed in the occurrence of diabetes and its vascular complications from 29% to 58% [126,127,128].

Last but not least, recent studies have indicated that DR is not only a microvascular disease but may be a result of neurodegenerative retinal changes. There is evidence of a correlation between retinal neurodegeneration and vascular abnormalities in diabetic patients. Functional and structural detection of the early stages of diabetic retinopathy may precede microvascular changes of diabetic retinopathy. Retinal neurodegeneration may be progressive and is independent of glycated hemoglobin, age, and sex [129,130,131].

## Figures and Tables

**Figure 1 jcm-10-00458-f001:**
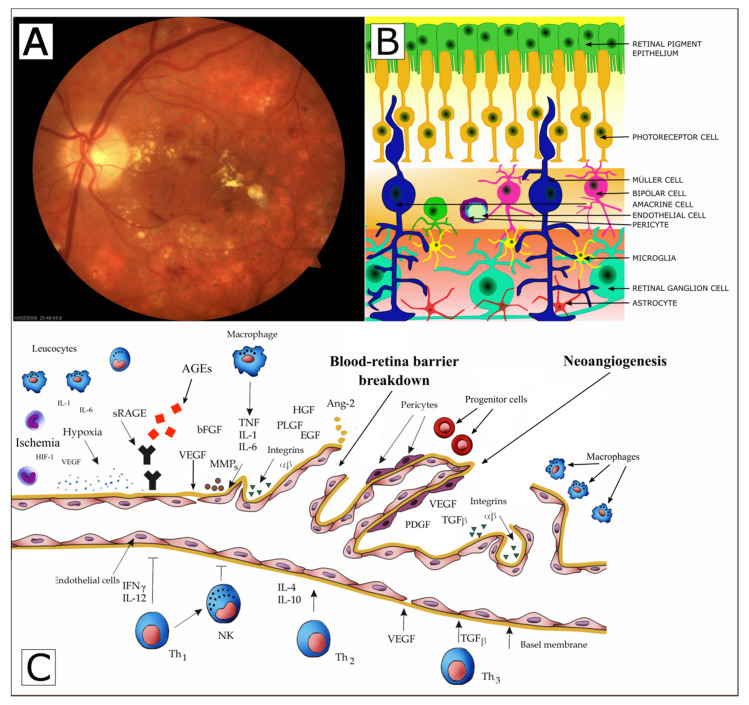
Possible mechanism of retinal vascular endothelial cell dysfunction and neuroretinal degeneration in diabetic patients. Fundus photos of eyes from diabetic retinopathy accompanied by macular oedema (**Panel**
**A**), modified from [42]. The neurovascular unit of the retina (retinal ganglion cells, Müller cells, microglia, astrocytes, endothelial cells, pericytes, and other). (**Panel**
**B**)**,** modified from [43]. The selected factors involved in the development and progression of diabetic retinopathy. AGEs—advanced glycation end products, RAGEs—receptor for advanced glycation end products, VEGF- vascular endothelial growth factor, IGF-I—insulin like growth factor, PLGF—placental growth factor, HGF—hepatocyte growth factor, PEDF pigment epithelium derived factor, bFGF—basic fibroblast growth factor, TGF-beta. transforming growth factor beta, MMPs—metalloproteinases PDGF-platelet-derived growth factor, EGF-Epidermal growth factor, Ang-2—Angiopoietin-2, (**Panel**
**C**), modified from [42,43].

## Data Availability

Data sharing not applicable.

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
