# Peer review of "Retinal Vascular Endothelial Cell Dysfunction and Neuroretinal Degeneration in Diabetic Patients"

_jcm, 2021, doi:10.3390/jcm10030458_

Round 1

Reviewer 1 Report

This paper is somewhat improved and there is some hope that it will eventually be a reasonable contribution.  However, it is not there yet.  There are numerous places where what the authors are trying to say is not clear, and there are redundancies, contradictions, missing references, and mistakes. It is still mainly a collection of facts without paying much attention to what species was used for different papers, and little synthesis of the different ideas into a coherent story. It is not carefully written, and this it is not just a question of the English.  However, there are problems with the English also, and I do not think it can be published without a thorough review for scientific style by a native English speaker.  I have tried to comment on many of the problems, but I am not sure I have caught them all.

  1. 13 “pericyte loss” or “loss of pericytes,” not “pericytes loss.”
  2. 23, 24. This should say something like: “Changes in the function and the structure of neural and glial cells in early DR may precede…”  The sentence currently does not say what function or structure may precede vascular changes.
  3. 36-37. Should be “As many as 4.7 million patients died due to DM complications.”  And is this the number who died in one year?  In the US?
  4. 42. “The time of lesions progression…” should be just “Lesion progression…”
  5. 44, 45. For some reason, the terms type 1 and type 2 were replaced by abbreviations, but the definition of those abbreviations is not given.  DR is also not defined the first time it is used (in abstract and in line 44)
  6. 54-57. Two successive sentences say practically the same thing.
  7. 54, 58. There are other basement membranes besides those associated with blood vessels, so this needs to be more specific.
  8. 70: What is the change that is observed at one week? Also, there are a number of changes that may occur early, just due to general inflammation, and they resolve and are not really signs of retinopathy.  If the authors look up the early changes in VEGF, they will find that in rats there are changes in the time frame of around 1 month, but later, these disappear, and of course rats never show neovascularization, so it seems that some of these early changes are not sustained and are not actually relevant to the disease process affecting either neural or vascular cells.
  9. 72 and other places. “organ of sight” and “organ of site cells” are odd terms.  Why not say eye or retina, as appropriate?  If the cells being referred to are a specific class, it should be given; if not, just say retinal cells.
  10. 88-91: It is not clear if IL-34 is better than CRP for discriminating retinopathy or other complications.  If the paper referred to (24) is not about retina, I am not sure that these sentences are relevant.  The complications in different organs are somewhat different, although many involve the vasculature, so this whole section 2 needs to be specific about what complications are being discussed.
  11. 104-105 should be “healthy children”
  12. 116, 117 – To me, it does not seem reasonable that one could sample aqueous humor to assess DR progression or monitor treatment. The authors of the article you cite may suggest that, but do you think this is reasonable?
  13. 128: TNFalpha turned out to be the “paramount tested factor” for what?
  14. 130: NPDR should not be in parentheses
  15. 132 -136: Is TNFalpha predictive of NPDR or of PDR, or both? In line 136, are you referring to a marker, or is TNFalpha a marker of the inflammatory state?
  16. 142: should be “compared to patients without PDR.”
  17. 143: “In PDR patients compared to PDR ones”? Something is wrong here.
  18. 159: Simply putting the figure in the paper is not particularly useful unless you describe it and discuss steps in the process of neovascularization. Some of these steps are discussed, but not all, and not with respect to this figure.
  19. 160-165: There is a big assumption that HIF-1 always implies hypoxia, but first, what is the evidence that HIF-1alpha is increased in DR (and at what stage), and second, what is the evidence that there is hypoxia? None of the references are about that point, and Joussen et al, in particular, is pretty early diabetes, where there is not much change in any histological or physiological parameter.  Certainly after there is capillary loss there must be hypoxia, but evidence of hypoxia at earlier times in DR, driving the process, or hyperglycemia directly causing hypoxia are not, to my knowledge, established.  If the authors can come up with references on these points, it would be useful.  Otherwise, they need to be more cautious about implicating hypoxia as causal.  VEGF can increase for many reasons, not just HIF upregulation. Other good reviews are careful about making assumptions about hypoxia.
  20. 176, 181, 184: VEGF was detected where? In addition to the references given, there are two important references about VEGF increasing in background retinopathy: Amin et al IOVS 38:36-47 1997 and Kunz Mathews et al. IOVS 38:2729-2741 1997
  21. 186: do you mean “until complications are present”?
  22. 206; “led” if you mean past tense, “leads” if present tense.
  23. Figure 1B. On the figure, microglia and endothelial cell are misspelled.
  24. 263 – 268: The structural part of this is written in a confusing way. The basement membrane of endothelial cells is right outside the basal membrane. The pericytes are separated from epithelial cells by the basement membrane.  Astrocytes in the retina exist only in the nerve fiber layer, and Muller cells are also involved in the vascular-glial unit. Kur et al. Prog. Ret and Eye Res 31:377-406, 2012 gives an outline of the relation between glial cells and blood vessels, and the references in that article would be instructive.  The references given now in this section do not seem specific to the retinal circulation.
  25. 339-344: This section is confusing. What is angiogenic paradox?  Why in line 340 is CD34+ repeated? Is there a class of cells that is CD34 and KDR positive and the comma is not supposed to be there?  What is the functional type of these cells? Are these expressed in the retina?
  26. 350-352: the cytokines may be observed in the vitreous and aqueous, but “increasing adhesiveness of leukocytes” cannot be observed in the vitreous and aqueous, as the sentence reads now.
  27. 362: what is the reference for the statement “enhancement of glucose metabolism… is assisted by increased production of free oxygen radicals…” This may be correct, but does not seem correct to me.
  28. 363: I think you mean endothelium not epithelium
  29. 390: What accelerated apoptosis are you referring to? What is the reference for this?
  30. 392: I think you probably mean microglia, that is, immune system cells, not glia in this sentence. Muller cells are the only glia, apart from astrocytes, and they do not migrate.  
  31. 397-399: These two sentences are redundant.
  32. 410. “arterial wall” is not a type of cell, so it does not fit in this list. You must mean some particular type of cell in the arterial wall.
  33. 415-428: this section about ET-1 is quite confusing and needs to be rewritten. Line 415 says that there is enhanced function of ET-1, although the sentence itself is confusing. Do you mean “… reported that production and function of ET-1 are substantially altered…”  “Increased” and “enhanced” do not make sense.  “Increased” and “enhanced” with respect to what?  Then I think you mean, in lines 416 and 417: “…with two separate G proteins mediate the effect of ET-1…”  The G proteins do not affect ET-1, but the effect of ET-1.  Lines 419-420 is largely redundant with l. 413-414.  Line 421 refers to a fall in ET-1 production or action, and line 427 refers to ET-1 upregulation.
  34. Section 10. I am not convinced that a discussion about flow mediated dilatation belongs here, because, as you point out, this is determined with the brachial artery, and is relevant for large vessels, but not necessarily those in the retina. If you do leave this in, it requires several clarifications. First, if the pressure in the cuff (sphygmomanometer) is just over 50 mmHg, that will not occlude blood flow in the arm, so something is wrong here.  Also, the “systolic” pressure refers to pressure in the arm, not in the cuff.  In line 447, you should not start this sentence “In diabetic retinopathy…”  Just say: “The mechanisms of …differ from those in the retinal microcirculation.”  In line 449, what neurohormonal mechanism are you referring to?
  35. 458 – 468: This section does not have a single reference, and there are ideas here not discussed previously in the paper.
  36. 470-478: This section is very awkwardly written: “that they conditioned the neurons function change” would be improved if it were: “that they were responsible for the change in neuronal function.”  And in the next sentence, “neuronal function” would be better. Then “Diabetics also have abnormal electrophysiological tests in patients in which vascular lesions….” would be an improvement.   This statement about electrophysiological tests requires references.
  37. 479: do you mean the rim of the optic disk?
  38. 488-492: you have not referred to the molecules mentioned here (ephrins, etc) before, and this section needs references if others have implicated them in DR. If this is your suggestion based on some other degeneration, it still needs references and it needs to be clear what the rationale for these statements is.
  39. 498: you probably do not mean “no diabetes” but instead “no retinopathy”
  40. 532: While PEDF eye drops might work in a rodent with a very small eye, they are very unlikely to work in humans. I do not know of any topical eye treatment that is useful in any retinal disease.  The retina is just too far from the cornea in humans for topical treatments to reach the retina. I do not think you should believe this suggestion.
  41. 545 – 547: I do not understand this sentence.

Author Response

Response to Reviewer 1 Comments

Thank you for preparing the review of our manuscript ID: jcm-1035004. All your comments were taken into account.

Point 1.

13 “pericyte loss” or “loss of pericytes,” not “pericytes loss.”

Response 1:

Thank you for your remark. It was changed to “pericyte loss

Point 2.

 23, 24. This should say something like: “Changes in the function and the structure of neural and glial cells in early DR may precede…”  The sentence currently does not say what function or structure may precede vascular changes.

Response 2.

Thank you for your remark. We have made appropriate changes in the point.

Neuroretinal degeneration appears structurally, as neural apoptosis of amacrine and Muller cells, reactive gliosis, ganglion cell layer/inner plexiform (GCL) thickness, retinal thickness and retinal nerve fiber layer thickness, reduction of the neuroretinal rim in minimum rim width (MRW); and functionally, as abnormal electroretinogram (ERG), dark adapatation, contrast sensitivity, color vision, and microperimetric  test.

Point 3.

36-37. Should be “As many as 4.7 million patients died due to DM complications.”  And is this the number who died in one year?  In the US?

Response 3.

Thank you for your remark. We have made the correction in the point.

Globally, as many as 4.7 million patients died due to DM complications for the year 2012.

Point 4.

  1. “The time of lesions progression…” should be just “Lesion progression…”

Response 4.

It has been corrected as “Lesion progression…”

Point 5.

44, 45. For some reason, the terms type 1 and type 2 were replaced by abbreviations, but the definition of those abbreviations is not given.  DR is also not defined the first time it is used (in abstract and in line 44

Response 5.

Thank you for your remark. The definitions were supplemented.

type 1 and 2 diabetes mellitus (T1DM and T2DM, respectively); Abstract: Diabetes mellitus (DM)

Point 6.

54-57. Two successive sentences say practically the same thing.

Response 6.

 Thank you for your remark. The first sentence was removed.

When the time of duration of diabetes increases, substantial vascular lesions are more likely to affect the retina. As the duration of diabetes increases, the probability of remarkable vascular alterations in the retinal tissue rises.

Point 7.

54, 58. There are other basement membranes besides those associated with blood vessels, so this needs to be more specific.

Response 7.

Thank you for your remark. This point has been described in more details in the context of specificity of the basement membrane associated with blood microvessels.

Pericytes are the mural cells of blood microvessels, which have recently come into focus for modulating angiogenesis, regulating blood flow, and maintaining blood-retina barrier (BRB) integrity. Pericytes lying on the capillaries, and are surrounded by the basement membrane, they can prevent ischemia-reperfusion after thrombus clearance by constricting capillaries whereas their relaxation increases blood flow

Point 8.

70: What is the change that is observed at one week? Also, there are a number of changes that may occur early, just due to general inflammation, and they resolve and are not really signs of retinopathy.  If the authors look up the early changes in VEGF, they will find that in rats there are changes in the time frame of around 1 month, but later, these disappear, and of course rats never show neovascularization, so it seems that some of these early changes are not sustained and are not actually relevant to the disease process affecting either neural or vascular cells.

Answer 8.

 Thank you for your attention. In the revised version of the manuscript, we added the words, (blood-retinal barrier breakdown) and cited the reference accordingly [20].

It now reads: Studies conducted in non-obese diabetic mice (NOD mice) have demonstrated that the first changes on the fundus of the eye blood-retinal barrier breakdown are observed already in the first week of exposure to high glucose levels [20].

  1. Xu Q, Qaum T, Adamis AP. Sensitive blood-retinal barrier breakdown quantitation using Evans blue. Invest Ophthalmol Vis Sci. 2001 Mar;42(3):789-94.

Results….Quantitation of Diabetes-Induced, Blood–Retinal Barrier Breakdown Using Evans Blue

Figure 5 shows the retinal blood vessel leakage (μl plasma × g retinal dry wt−1 · h−1) of 1-week diabetic rats that received a single 60 mg/kg intraperitoneal injection of streptozotocin 1 week prior. Data for nondiabetic control rats, which received equivalent volume of citrate buffer only, is also displayed. The mean ± SE of retinal Evans blue leakage was 5.36 ± 0.80 (n = 10 retinas) and 9.28 ± 0.90 (n = 11 retinas) μl plasma × g retinal dry wt−1 · h−1 for nondiabetic and diabetic rats, respectively (P < 0.05). Thus, the retinal Evans blue leakage of 1-week diabetic rats was approximately 1.7-fold higher than that of nondiabetic control rats. [20]

Discussion ….”For this purpose, 1-week diabetic animals and age-matched nondiabetic controls were used. The results showed that the leakage of retinal Evans blue in diabetic animals is 1.7-fold higher than that in normal controls”…... [20]

Point 9.

72 and other places. “organ of sight” and “organ of site cells” are odd terms.  Why not say eye or retina, as appropriate?  If the cells being referred to are a specific class, it should be given; if not, just say retinal cells.

Answer 9.

Thank you for your remark. The term „Organ of sight” was changed to”the retinal cells”

Point 10.

88-91: It is not clear if IL-34 is better than CRP for discriminating retinopathy or other complications.  If the paper referred to (24) is not about retina, I am not sure that these sentences are relevant.  The complications in different organs are somewhat different, although many involve the vasculature, so this whole section 2 needs to be specific about what complications are being discussed.

Answer 10.

Thank you for asking this question.

In the publication [24], in the methodology section, the authors described in detail patients with T2DM and micro- and macrovascular complications. The T2DM group also comprised patients with diabetic retinopathy. In the cited work, the authors used strong statistical methods and presented the obtained results in a manuscript that was accepted for review and then published in the Biomarkers journal.

Point 11.

104-105 should be “healthy children”

Answer 11.

 Thank you for your remark. It was changed to “healthy children”

Point 12.

116, 117 – To me, it does not seem reasonable that one could sample aqueous humor to assess DR progression or monitor treatment. The authors of the article you cite may suggest that, but do you think this is reasonable?

Answer 12.

 Thank you for your remark.

To our knowledge, all research (both on animals and humans) is carried out according to strictly defined procedures and require the consent of the bioethics committee. I have read the cited publications once again and confirm that in both of the cited manuscripts the authors received the approval of the bioethics committee. Moreover, the methodology precisely defines the sampling procedure. And so, for example, in article [23] the authors wrote that: “aqueous humor sample (100 μl) was obtained before intravitreal anti-VEGF injection by anterior chamber paracentesis.”

  1. Song, S.; Yu, X.; Zhang, P. et al. Increasedlevels of cytokines in the aqueous humor correlate with the severity of diabetic retinopathy. J Diabetes Complications. 2020 Sep,34(9),107641

(below is part of the text of the cited manuscript)

This prospective study followed the principles of the Helsinki Declaration and was approved by the Hospital Ethics Committee. All participants signed a standard informed consent form.

This study included 103 patients diagnosed with DR secondary to type 2 diabetes mellitus and 103 eyes (1 eye per patient) that received intravitreal injection of ranibizumab (0.5 mg/0.05 ml) for the first time between September 2018 and August 2019. All patients were confirmed by ophthalmologists and examined in detail, including best corrected visual acuity (BCVA) and intraocular pressure (IOP), slit-lamp biomicroscopy, automatic optometry, gonioscopy, optical coherence tomography (OCT) and fundoscopic examinations of both eyes.

Patients who met the following criteria were excluded: (1) had other serious diabetic complications, such as nephropathy, ketoacidosis, or hyperosmotic coma; (2) had other ocular diseases or a history of ocular surgery including intravitreal injections; (3) had other major systemic diseases; and (4) had undergone retinal laser photocoagulation.

DR was classified according to the results of indirect ophthalmoscopy and OCT or FFA. The patients were categorized into 3 groups: nonproliferative diabetic retinopathy (NPDR) group (42 eyes), proliferative diabetic retinopathy (PDR) without neovascular glaucoma (NVG) group (40 eyes) and NVG-PDR group (21 eyes). NPDR subjects were patients with diabetic macular edema who needed intravitreal anti-VEGF injection. PDR subjects were patients with fibrovascular proliferation and vitreous hemorrhage who needed vitrectomy and intravitreal anti-VEGF injection within 7 days before vitrectomy. NVG subjects were patients with iris neovascularization and elevated intraocular pressure who needed intravitreal anti-VEGF injection.

2.2. Sample collection and detection

The undiluted aqueous humor sample (100 μl) was obtained before intravitreal anti-VEGF injection by anterior chamber paracentesis. The aqueous humor samples were immediately sealed and stored in a plastic tube (500 μl; Axygen Inc., USA) at −80 °C until assayed. The samples were assayed within 6 months after collection. All cytokines were analyzed by Becton Dickinson CBA software following the standard protocol. CBA technology has been described in detail before. Each assay of this study was performed as recommended by the manufacturer.

Point 13.

128: TNFalpha turned out to be the “paramount tested factor” for what?

Answer 13.

 Thank you for your attention. The sentence was corrected accordingly.

The fragment originally read: On the other hand, clinical studies in the group of type 1 diabetic children showed that TNFα turned out to be the paramount tested factor [37].

It now reads: On the other hand, clinical studies in the group of type 1 diabetic children showed that among the analyzed parameters TNF-alpha appears to be the most significant predictor of damage to the eye apparatus  [37].Point 14.

NPDR should not be in parentheses

Answer 14.

Parentheses were removed.

Point 15.

132 -136: Is TNFalpha predictive of NPDR or of PDR, or both? In line 136, are you referring to a marker, or is TNFalpha a marker of the inflammatory state?

Answer 15.

 Thank you for this question.

In 2007,  Zorena et al. showed that “Serum TNF-alpha level predicts nonproliferative diabetic retinopathy (NPDR) in children.” Mediators Inflamm. 2007:92196 [36].

A few months later Gustavsson, C. et al. showed that “TNF-alpha is an independent serum marker for proliferative retinopathy (PDR)  in adult patients with type 1 diabetes.”  J Diabetes Complications.2008 Sep-Oct,22(5),309-16 [37].

The answer: TNF alpha is a predictor of NPDR and PDR in patients with type 1 diabetes

Point 16.

142: should be “compared to patients without PDR.”

Answer 16.

Thank you for your remark. It is now “compared to patients without PDR.”

Point 17.

143: “In PDR patients compared to PDR ones”? Something is wrong here.

Answer 17.

 Thank you for all invaluable remarks.

The words "active PDR" "inactive PDR" have been clarified/added.

It should be: „patients with active PDR” and the second group: „patients with inactive PDR”

The authors edited the manuscript accordingly and additionally wrote:

The activity of disease was defined based on ophthalmologic evaluation (preoperative and intraoperative). The activity of the disease was categorized as active PDR when there were perfused capillaries in neovascular membranes, or as inactive or quiescent PDR when no vessels could be discerned in fibrotic membranes or when no blood could be seen in gliotic vessels in fibrotic membranes [40].

  1. Urbančič, M.;Petrovič, D.;Živin, A.M. et al. Correlations between vitreous cytokine levels and inflammatory cells in fibrovascular membranes of patients with proliferative diabetic retinopathy. Mol Vis.2020 Jun 26;26:472-482

Point 18.

159: Simply putting the figure in the paper is not particularly useful unless you describe it and discuss steps in the process of neovascularization. Some of these steps are discussed, but not all, and not with respect to this figure.

Answer 18.

 Thank you for pointing this out.

We added the following:

The first ophthalmoscopic symptoms in the course of diabetes mellitus include microfibromas or focal ecchymoses in the macular area and retinal edema.  Then, on the border of the normal and swollen retina, lipids are deposited and the formation of "hard exudates" occurs. They appear at the back of the eye in the form of dots, spots or plaques Figure 1, (Panel A) [43].

…..factors such as interleukin 1 (IL1), tumor necrosis factor α (TNF-α) and vascular endothelial growth factor (VEGF), adhesive molecules, and the activation of a nuclear transcription factor NFκB, mediating pericyte apoptosis, vascular inflammation and angiogenesis, as well as breakdown of the inner blood-retinal barrier (BRB), the end result of all these events is damage to the neural and vascular components of the retina Figure1 (Panel C) [18, 43-44].

Point 19.

160-165: There is a big assumption that HIF-1 always implies hypoxia, but first, what is the evidence that HIF-1alpha is increased in DR (and at what stage), and second, what is the evidence that there is hypoxia? None of the references are about that point, and Joussen et al, in particular, is pretty early diabetes, where there is not much change in any histological or physiological parameter.  Certainly after there is capillary loss there must be hypoxia, but evidence of hypoxia at earlier times in DR, driving the process, or hyperglycemia directly causing hypoxia are not, to my knowledge, established.  If the authors can come up with references on these points, it would be useful.  Otherwise, they need to be more cautious about implicating hypoxia as causal.  VEGF can increase for many reasons, not just HIF upregulation. Other good reviews are careful about making assumptions about hypoxia.

Answer 19.

Thank you for your remark.

It is now: In the course of studies it has been demonstrated that hyperglycaemia leads to hypoxia and inflammation. The hyperglycaemia elevates hypoxia induced factor 1 (HIF-1) and insulin-like growth factor 1 (IGF-1), both in the serum and the vitreous body of diabetic patients [49-51].

In three selected references [49-51], the authors emphasize the relationship between hyperglycaemia, hypoxia and inflammation, and so manuscript "49" is about hyperglycemia in DR,

  1. Liu, H.; Lessieur, E.M.; Saadane, A. et al. Neutrophil elastase contributes to the pathological vascular permeability characteristic of diabetic retinopathy. Diabetologia. 2019 Dec,62(12),2365-2374.

manuscript "50" is about hypoxia and inflammation in DR,

  1. Gaonkar, B.; Prabhu, K.; Rao, P. et al. Plasma angiogenesis and oxidative stress markers in patients with diabetic retinopathy. Biomarkers. 2020 Jul,25(5),397-401.

and an excellent manuscript “51” Poulaki, V.; Joussen, A.M.; Mitsiades, N. et al. Insulin-like growth factor-I plays a pathogenetic role in diabetic retinopathy.  Am J Pathol. 2004 Aug,165(2),457-69. The authors conducted research on both human RPE cells as well as nondiabetic and diabetic rats. They presented the mechanism of diabetic retinopathy in which hypoxia induced factor 1 (HIF-1) is involved [51].

Point 20.

176, 181, 184: VEGF was detected where? In addition to the references given, there are two important references about VEGF increasing in background retinopathy: Amin et al IOVS 38:36-47 1997 and Kunz Mathews et al. IOVS 38:2729-2741 1997

Answer 20.

 Thank you for your remark. The paragraph has been corrected and the references were added.

VEGF expression precedes retinal neovascularization in the retinas and the optic nerves of humans with diabetes. Its localization to glial cells of the inner retina and the anterior optic nerve suggests a relationship to neovascularization in these sites. That VEGF immunopositivity may occur when there is no morphological evidence of retinal nonperfusion and little likelihood of retinal neovascularization suggests the possibility that ischemia may not be the only stimulus for VEGF expression [55]. ]. In addition, VEGF immunoreactivity is correlated with increased vascular permeability, as indicated by human serum albumin (HSA) immunostaining and appears to be increased in diabetic subjects before the onset of retinopathy [56].

  1. Amin, R.H.; Frank, R.N.; Kennedy, A. et al. Vascular endothelial growth factor is present in glial cells of the retina and optic nerve of human subjects with nonproliferative diabetic retinopathy. Invest Ophthalmol Vis Sci. 1997 Jan;38(1):36-47. 
  2. Mathews, M.K.; Merges, C.; McLeod, D.S. et al. Vascular endothelial growth factor and vascular permeability changes in human diabetic retinopathy. Invest Ophthalmol Vis Sci. 1997 Dec;38(13):2729-41.

Point 21.

186: do you mean “until complications are present”?

Answer 21.

Thank you for your attention. The word "vascular" was  added.

It now reads: These data indicate that until no vascular complications are present in T1DM patients, VEGF level is not significantly higher in comparison with the healthy control group.

Point 22.

206; “led” if you mean past tense, “leads” if present tense.

Answer 22.

 Thank you for your remark. It is now ”leads”.

Point 23.

Figure 1B. On the figure, microglia and endothelial cell are misspelled.

Answer 23.

 Thank you for your attention. Figure 1B has been corrected accordingly.

It currently reads: microglia, endothelial cells.

Point 24.

263 – 268: The structural part of this is written in a confusing way. The basement membrane of endothelial cells is right outside the basal membrane. The pericytes are separated from epithelial cells by the basement membrane.  Astrocytes in the retina exist only in the nerve fiber layer, and Muller cells are also involved in the vascular-glial unit. Kur et al. Prog. Ret and Eye Res 31:377-406, 2012 gives an outline of the relation between glial cells and blood vessels, and the references in that article would be instructive.  The references given now in this section do not seem specific to the retinal circulation.

Answer 24.

 Thank you for your remark. This paragraph has been deleted and appropriate and new information has been included in  paragraph No 4.

  1. The neurovacular unit of the retina

The retinal neurovascular unit includes the physical and biochemical relationship among neurons, glia, and blood vessels and the close interdependency of these tissues in the retina and the central nervous system. The neural unit (ganglion cells and glial cells) and the vascular unit (endothelial cells and pericyte) constitute the retinal neurovascular unit, together with retinal pigment epithelial (RPE) cells, they are the main body of BRB. The BRB is divided into two parts: the inner blood-retina barrier (iBRB) consists of retinal endothelial cells that are covered by astrocytes, pericytes, and Müller cells end-foot and is essential for maintaining the microenvironment of the inner layers of the retina; the outer blood-retina barrier (oBRB) is composed of tight junctions formed by neighboring RPE cells and serves as a filter to regulate solutes and nutrients from the blood. Hyperglycemia causes cells dysfunction, when impairment occurs, structural defect and functional disorder arise in the neurovascular unit, which in turn lead to further cells impairment [13, 62-64]. 

4.1. The neural unit

The neural unit includes retinal ganglion cells(RGCs), glial cells, and other neural cell types.

4.1.1. RGCs

The Integrity of RGCs provides the normal function of the retina, however, abnormal environment such as hyperglycemia alters the structure or function of RGCs and this kind of RGCs impairment is progressive in the development of subsequent DR.

4.1.2. Glial cells

There are three major types of glial cells in the retina and they are involved in maintaining retinal homeostasis: Müller cells, astrocytes, and microglia. 

Müller cells are the leading type of glial element, representing 90 % of the retinal glia. The inner limiting membrane is made up by the bases of Müller cells and connects to the base of the vitreous humor. The circular junctions between photoreceptors and Müller glial cells form the outer limiting membrane [61]. Müller cells manage vascular responses to fulfill the metabolic demand of neurons, interchange metabolites, recycle neurotransmitters, and aid establishing the extracellular chemical environments [65]. It has been proved that the population of these cells shows traits of stem cells and is able to develop in different cells of the retina. The accelerated apoptosis of ganglion cells is accompanied by lesions within Müller glial cells. 

Hyperglycemia and inflammation induce activation of microglial cells that are found in the inner part of the retina and migrate later to the subretinal space and release cytokines, thus causing death of nerve cells. The activated microglial cells adhere to the vessels and seem to play a crucial role in vascular wall damage [66,67]. Similar results were obtained by Schellini et al., who investigated the morphology of Müller cells in the retina of healthy rats, with treated and untreated diabetes, 1 and 12 months following the induction of diabetes. In treated rats the lesions were less distinct [68-70]. It was also found that Müller cells in diabetic patients showed increased expression of glial acidic fibrilar protein (GFAP) [71]. Since Müller cells produce factors able to modulate blood flow, vascular permeability and cell survival, they are thought to have a major effect on endothelial function.

The connection between Müller cells and the vascular unit is affected during DR and they produce large amounts of factors, while some show protective effect, more are making it worse such as vascular endothelial growth factor(VEGF). VEGF is upregulated in the early stages of DR and presents a neuroprotective effect for the maintenance and survival of neurons, VEGF derived from Müller cells in diabetes can also increase blood circulation for proper metabolic activities [55-56].

Named based on their stellate shape, astrocytes are linked to the retinal blood vessels and neurons and play a significant role in the BRB integrity. In contrast to Müller cells, astrocytes are considered almost exclusively located in the innermost retinal layers. The distribution of astrocytes is related to the presence and position of retinal blood vessels, therefore, astrocytes are more likely to locate in the vascular regions of the retina, while avascular zones contain nearly no astrocytes [72]. Under the diabetic environment, characterized by dysmetabolic and hypoxic, astrocytes are activated and produce a variety of pro-inflammatory cytokines as well as chemokines, together with T-cells, monocytes/macrophages, and microglia, amplifying the inflammatory response [73]. 

As the resident inflammatory cells of the retina, in addition to immunologic activity, microglia also participate in neural network development and maintenance. In non-proliferative DR(NPDR), the increased perivascular microglia cells settled into the retinal plexiform layers and exhibit mildly hypertrophic. In PDR, microglial cells gathered in ischemic areas and new dilated blood vessels heavily. The contact between Microglia and RGCs are also found in epiretinal membranes. These findings suggest that activated microglia involved in all stages of DR and may even accelerate DR progression [74].

4.2. The vascular unit

Micro-vasculopathy is a typical characteristic of DM, which can eventually lead to vision loss. As the basic unit of retinal vascular structure and function, the stability of pericytes and endothelial cells is necessary for maintaining vascular integrity.

4.2.1. Endothelial cells (EC)

Endothelial cells (EC) (monolayer squamous epithelium) form the smooth internal vascular lining indispensable for normal blood flow. They play a major role in cardiovascular homeostasis maintenance, mainly through the regulation of arterial blood pressure. Apart from ensuring the barrier between the vessels and its lumen, endothelial cells secrete numerous mediators regulating platelet aggregation, fibrinolysis, coagulation and the tension of the vessels. Endothelial dysfunction observed in diabetic patients contributes to the development of lesions in large and small vessels. The major factors include hyperglycemia, insulin resistance, hyperinsulinemia and dyslipidemia. Endothelial cells secrete a few mediators that can alternatively mediate in vascular stenosis [75-77]. The complex tight junctions of ECs are one of the significant parts of the iBRB, which protects the retina from injuries by circulating substances or metabolites. Under the development of DR, increased capillary occlusion results in retinal ischemia and hypoxia, triggering an increase in the levels of VEGF, which then promotes pathological neovascularization. The receptors on ECs are activated by VEGF, and this kind of activation promotes ECs survival, proliferation, permeability, and taxis toward the VEGF sources, the essential steps of blood vessels production and maintenance, however, in many DR patients, VEGF is excessively produced in the retina resulting in the over-proliferation of ECs, new blood vessel formation, thus contributing to microaneurysms, fluid leakage, and tissue damage [78].

Endothelial progenitor cells (EPC) are immature cells able to differentiate into mature endothelail cells. Vascular damage and tissue ischemia mobilize the EPC by releasing growth factors and cytokines. The EPC stimulate compensatory angiogenesis chiefly at the sites of ischemia in the peripheral circulation. Furthermore, the EPC make up a circulating pool of cells capable of forming a “cellular patch” at sites of endothelial damage, simultaneously contributing directly to homeostasis and repair of the endothelial layer.  Tissue ischemia is regarded as the most powerful stimulus of EPC mobilization by means of activation of the systems that detect hypoxia, such as hypoxia-induced factor (HIF) -1α. Both patients with type 1 and 2 diabetes have fewer circulating EPC than healthy individuals. EPC in diabetic patients show the impairment of the function, such as decreased proliferation, adhesion, migration and a potential to be incorporated into vascular structures  [13, 76-77].

4.2.2. Pericytes 

Pericytes are the mural cells of blood microvessels, which have recently come into focus for modulating angiogenesis and maintaining BRB integrity [79]. Pericytes lying on the capillaries and are separated from epithelial cells by the basement membrane. They can prevent ischemia-reperfusion after thrombus clearance by constricting capillaries whereas their relaxation increases blood flow [80].  

Loss of pericytes, accompanied by ECs degeneration and the basement membrane thickening, may leads to the formation of abnormal capillaries which can generate microaneurysm, and are related to vascular occlusion, or induced by thrombosis secondary to residual ECs impairment, causing ischaemic-hypoxic retinal damage [81].

The references given before in this section do not seem specific to the retinal circulation were removed, and relevant reference suggested by Reviewer, was included.

  1. Kur, J.; Newman, E.A.; Chan-Ling, T. Cellular and physiological mechanisms underlying blood flow regulation in the retina and choroid in health and disease. Prog Retin Eye Res. 2012 Sep,31(5),377-406. 

Point 25.

339-344: This section is confusing. What is angiogenic paradox?  Why in line 340 is CD34+ repeated? Is there a class of cells that is CD34 and KDR positive and the comma is not supposed to be there?  What is the functional type of these cells? Are these expressed in the retina?

Answer 25.

 Thank you for your remark. This section has been deleted because these cells do not play important role in this process.

Point 26.

350-352: the cytokines may be observed in the vitreous and aqueous, but “increasing adhesiveness of leukocytes” cannot be observed in the vitreous and aqueous, as the sentence reads now.

Answer 26.

 Thank you for your remark. This sentence was corrected

Additionally, higher levels of IL-6, IL-7 and IL-8, that cause increasing adhesiveness of leucocytes to vascular endothelium, have been observed in the vitreous and aqueous humour. It is considered that chronic inflammation of the vascular walls also encourages their occlusion and intensifies retinal ischemia [89-90].

Point 27.

362: what is the reference for the statement “enhancement of glucose metabolism… is assisted by increased production of free oxygen radicals…” This may be correct, but does not seem correct to me.

Answer 27.

 Thank your for your attention.

 Lee, W.C.; Mokhtar, S.S.; Munisamy, S. et a. Vitamin D status and oxidative stress in diabetes mellitus. Cell Mol Biol (Noisy-le-grand). 2018 May 30;64(7):60-69.

Point 28.

363: I think you mean endothelium not epithelium

Answer 28.

Thank you for your remark. You are right. It is now: endothelium

Point 29.

390: What accelerated apoptosis are you referring to? What is the reference for this?

Answer 29.

 Thank you for your remark. This paragraph was rejected.

Point 30.

392: I think you probably mean microglia, that is, immune system cells, not glia in this sentence. Muller cells are the only glia, apart from astrocytes, and they do not migrate.  

Answer 30.

 Thank you for your remark. The latest data regarding microglia, Muller cells and astrocytes were included in Section 4.

4.1. The neural unit

The neural unit includes retinal ganglion cells(RGCs), glial cells, and other neural cell types.

4.1.1. RGCs

The Integrity of RGCs provides the normal function of the retina, however, abnormal environment such as hyperglycemia alters the structure or function of RGCs and this kind of RGCs impairment is progressive in the development of subsequent DR.

4.1.2. Glial cells

There are three major types of glial cells in the retina and they are involved in maintaining retinal homeostasis: Müller cells, astrocytes, and microglia. 

Müller cells are the leading type of glial element, representing 90 % of the retinal glia. The inner limiting membrane is made up by the bases of Müller cells and connects to the base of the vitreous humor. The circular junctions between photoreceptors and Müller glial cells form the outer limiting membrane [61]. Müller cells manage vascular responses to fulfill the metabolic demand of neurons, interchange metabolites, recycle neurotransmitters, and aid establishing the extracellular chemical environments [65]. It has been proved that the population of these cells shows traits of stem cells and is able to develop in different cells of the retina. The accelerated apoptosis of ganglion cells is accompanied by lesions within Müller glial cells. 

Hyperglycemia and inflammation induce activation of microglial cells that are found in the inner part of the retina and migrate later to the subretinal space and release cytokines, thus causing death of nerve cells. The activated microglial cells adhere to the vessels and seem to play a crucial role in vascular wall damage [66,67]. Similar results were obtained by Schellini et al., who investigated the morphology of Müller cells in the retina of healthy rats, with treated and untreated diabetes, 1 and 12 months following the induction of diabetes. In treated rats the lesions were less distinct [68-70]. It was also found that Müller cells in diabetic patients showed increased expression of glial acidic fibrilar protein (GFAP) [71]. Since Müller cells produce factors able to modulate blood flow, vascular permeability and cell survival, they are thought to have a major effect on endothelial function.

The connection between Müller cells and the vascular unit is affected during DR and they produce large amounts of factors, while some show protective effect, more are making it worse such as vascular endothelial growth factor(VEGF). VEGF is upregulated in the early stages of DR and presents a neuroprotective effect for the maintenance and survival of neurons, VEGF derived from Müller cells in diabetes can also increase blood circulation for proper metabolic activities [55-56].

Named based on their stellate shape, astrocytes are linked to the retinal blood vessels and neurons and play a significant role in the BRB integrity. In contrast to Müller cells, astrocytes are considered almost exclusively located in the innermost retinal layers. The distribution of astrocytes is related to the presence and position of retinal blood vessels, therefore, astrocytes are more likely to locate in the vascular regions of the retina, while avascular zones contain nearly no astrocytes [72]. Under the diabetic environment, characterized by dysmetabolic and hypoxic, astrocytes are activated and produce a variety of pro-inflammatory cytokines as well as chemokines, together with T-cells, monocytes/macrophages, and microglia, amplifying the inflammatory response [73]. 

As the resident inflammatory cells of the retina, in addition to immunologic activity, microglia also participate in neural network development and maintenance. In non-proliferative DR(NPDR), the increased perivascular microglia cells settled into the retinal plexiform layers and exhibit mildly hypertrophic. In PDR, microglial cells gathered in ischemic areas and new dilated blood vessels heavily. The contact between Microglia and RGCs are also found in epiretinal membranes. These findings suggest that activated microglia involved in all stages of DR and may even accelerate DR progression [74].

Point 31.

397-399: These two sentences are redundant.

Answer 31.

 Thank you for your remark. The sentences were rejected.

In addition, the changes within mtDNA caused by   reactive oxygen species (ROS) leads to mutagenesis in the mitochondria and surrounding structures, which are in charge of the production of endogenous reactive oxygen.  This is a cause to decreased production of energy below the level required for tissue functioning, which results in tissue dysfunction or even premature death

Point 32.

  1. “arterial wall” is not a type of cell, so it does not fit in this list. You must mean some particular type of cell in the arterial wall.

Answer 32.

Thank you for your attention. The sense of this sentence was changed.

ET-1 is the major cardiovascular isoform of the ET system produced mainly in endothelium, even though it can be generated in vascular smooth muscle cells (VSMC) of the arterial wall, macrophages, leukocytes, cardiomyocytes and fibroblasts too.

Point 33.

415-428: this section about ET-1 is quite confusing and needs to be rewritten. Line 415 says that there is enhanced function of ET-1, although the sentence itself is confusing. Do you mean “… reported that production and function of ET-1 are substantially altered…”  “Increased” and “enhanced” do not make sense.  “Increased” and “enhanced” with respect to what?  Then I think you mean, in lines 416 and 417: “…with two separate G proteins mediate the effect of ET-1…”  The G proteins do not affect ET-1, but the effect of ET-1.  Lines 419-420 is largely redundant with l. 413-414.  Line 421 refers to a fall in ET-1 production or action, and line 427 refers to ET-1 upregulation.

Answer 33.

Thank you for your remark. This section was corrected.

 Overproduction and increased functional effects of ET-1 are reported to be strongly altered in diabetic conditions. The ETA and ETB receptors, coupled with two distinct G-proteins, mediate the action of ET-1 on the vascular tone. The two receptors in combination with two separate G proteins affect ET-1 on vascular tension.

Point 34.

Section 10. I am not convinced that a discussion about flow mediated dilatation belongs here, because, as you point out, this is determined with the brachial artery, and is relevant for large vessels, but not necessarily those in the retina. If you do leave this in, it requires several clarifications. First, if the pressure in the cuff (sphygmomanometer) is just over 50 mmHg, that will not occlude blood flow in the arm, so something is wrong here.  Also, the “systolic” pressure refers to pressure in the arm, not in the cuff.  In line 447, you should not start this sentence “In diabetic retinopathy…”  Just say: “The mechanisms of …differ from those in the retinal microcirculation.”  In line 449, what neurohormonal mechanism are you referring to?

Answer  34.

 Thank you for your remark. This section has been removed.

Point 35.

458 – 468: This section does not have a single reference, and there are ideas here not discussed previously in the paper.

Answer 35.

 Thank you for your remark. This section has been deleted.

Point 36.

470-478: This section is very awkwardly written: “that they conditioned the neurons function change” would be improved if it were: “that they were responsible for the change in neuronal function.”  And in the next sentence, “neuronal function” would be better. Then “Diabetics also have abnormal electrophysiological tests in patients in which vascular lesions….” would be an improvement.   This statement about electrophysiological tests requires references.

Answer 36.

 Thank you for your remark. This section was rewritten.

While vascular retinal lesions in the diabetes process have been explored well, lesions in nerve cells of the retina are still under clinical research. Until recently, the majority of researchers were focusing on vascular retinal lesions with the presumption that they were responsible for the change in neuronal function conditioned the neurons function change. In recent times, findings have been suggesting that the changes of the neuronal function and their viability contribute to the formation of DR pathogenic mechanisms starting soon after diabetes appearance [97-100]. Diabetes induces neural apoptosis of ganglion, amacrine, and Muller cells, activation of microglia caused by chronic glutamate toxity, inflammatory glial activation, ganglion cell layer/inner plexiform (GCL) thickness, retinal thickness and retinal nerve fiber layer thickness [100-106]. In his studies, Piona et al. [107] reported that the reduction of the neuroretinal rim in minimum rim width (MRW) which is the shortest distance between Bruch’s membrane opening (BMO) and internal limiting membrane (ILM), may be a potential early marker of retinal degeneration detectable in T1DM-children. A relationship between MRW and mean HbA1c suggests that the control of glucose metabolism may affect an early neurodegeneration of the retina, starting from childhood. These retinal findings cause functional changes, which precede clinical DR and may occur prior to the diagnosis of DM. Functionally changes include deficits in pattern electroretinogram (pattern ERG), increased implicit timed in the multifocal ERG (mfERG), abnormal dark adaptation, abnormal contrast sensitivity [108-115]. Recent years have delivered numerous proofs supporting the hypothesis that DR is a neurodegenerative complication [116-118]. The latest 8-year observations have revealed that the reduction in the thickness of the internal retinal layer takes place in T1DM patients along with the course of the disease but still before the DR manifestation. This confirms the hypothesis of retinal neurodegeneration  [119].

The references regarding electrophysiological tests were added.

  1. Pescosolido, N.; Barbato, A.; Stefanucci, A. et al. Role of Electrophysiology in the Early Diagnosis and Follow-Up of Diabetic Retinopathy. J Diabetes Res. 2015;2015:319692.
  2. Han, Y.; Adams, A.J.; Bearse, M.A. Jr., et al. Multifocal electroretinogram and short-wavelength automated perimetry measures in diabetic eyes with little or no retinopathy. Arch Ophthalmol. 2004 Dec; 122(12):1809–15.
  3. Han, Y.; Bearse, M.A. Jr.; Schneck, M.E. et al. Mulltifocal electroretinogram delays predict sites of subsequent diabetic retinopathy. Invest Ophthalmol Vis Sci. 2004 Mar; 45(3):948–54.
  4. Pardue, M.T.; Barnes, C.S.; Kim, .MK. et al. Rodent Hyperglycemia-Induced Inner Retinal Deficits are Mirrored in Human Diabetes. Translational vision science & technology. 20143(3):6.
  5. Aung, M.H.; Kim, M.K.; Olson, D.E. et al. Early visual deficits in streptozotocininduced diabetic long evans rats. Invest Ophthalmol Vis Sci. 2013 Feb 15; 54(2):1370–7.
  6. Drasdo, N.; Chiti, Z.; Owens, D.R. et al. Effect of darkness on inner retinal hypoxia in diabetes. Lancet. 2002; 359(9325):2251–3.
  7. Dosso, A.A.; Yenice-Ustun, F.; Sommerhalder, J. et al. Contrast sensitivity in obese dyslipidemic patients with insulin resistance. Arch Ophthalmol. 1998 Oct; 116(10):1316–20.
  8. Sokol, S.; Moskowitz, A.; Skarf, B. et al. Contrast sensitivity in diabetics with and without background retinopathy. Arch Ophthalmol. 1985 Jan; 103(1):51–4

Point 37.

 479: do you mean the rim of the optic disk?

Answer 37.

 Thank you for your remark. The rim refers to the neuroretinal rim in minimum rim width (MRW)

In his studies, Piona et al. [107] reported that the reduction of the neuroretinal rim in minimum rim width (MRW) which is the shortest distance between Bruch’s membrane opening (BMO) and internal limiting membrane (ILM), may be a potential early marker of retinal degeneration detectable in T1DM-children. A relationship between MRW and mean HbA1c suggests that the control of glucose metabolism may affect an early neurodegeneration of the retina, starting from childhood.

Point 38.

488-492: you have not referred to the molecules mentioned here (ephrins, etc) before, and this section needs references if others have implicated them in DR. If this is your suggestion based on some other degeneration, it still needs references and it needs to be clear what the rationale for these statements is.

Answer 38.

 Thank you for your attention. This paragraph was corrected. The sentences connected with other degenerations were removed.

The mediators engaged in the link between neuroretinal degeneration and  microvascular changes have not been recognized yet. It is known that VEGF is significant in the early stages of DR. On the other hand, recently, a growing interest has been concentrated on the role of some other important molecules, such as ephrins, semaphorins, and netrin which are released early by damaged neurons and may stimulate the development of DR [120-122]. The before mentioned molecules are highly expressed in the retina and vitreous in patients with advanced DR.

The proper references were added.

  1. Kociok, N.; Crespo-Garcia, S.; Liang, Y. et al. Lack of Netrin-4 modulates pathologic neovascularization in the eye. Sci Rep. 2016;6(1):1–13.
  2. Moran, E. P.; Wang, Z.; Chen, J. et al. Neurovascular cross talk in diabetic retinopathy: pathophysiological roles and therapeutic implications. Am J Physiol Heart Circ Physiol.2016, 311, 738–749.
  3. Li, Y.; Chen, D.; Sun, L. et al. Induced Expression of VEGFC, ANGPT, and EFNB2 and Their Receptors Characterizes Neovascularization in Proliferative Diabetic Retinopathy. Ophthalmol. Vis. Sci. 2019 Oct 1;60(13):4084-4096. 

Point 39.

 498: you probably do not mean “no diabetes” but instead “no retinopathy”

Answer 39.

 Thank you for your attention. It was changed.

Point 40.

532: While PEDF eye drops might work in a rodent with a very small eye, they are very unlikely to work in humans. I do not know of any topical eye treatment that is useful in any retinal disease.  The retina is just too far from the cornea in humans for topical treatments to reach the retina. I do not think you should believe this suggestion.

Answer 40.

  •  

Thank you for your interesting point of view. I am more optimistic and I present the  promising examples.

1). Scientists at the University of Birmingham are one step closer to developing an eye drop that could revolutionise treatment for age-related macular degeneration (AMD). AMD is the leading cause of blindness in the developed world. Its prevalence is increasing dramatically as the population ages and it is estimated that, by 2020, there will be about 200 million people worldwide with the condition. In the UK alone, there are over 500,000 people with late-stage AMD.

AMD is currently treated by injections of sight-saving drugs into the eye which must be administered by medical professionals. Scientists led by biochemist Dr Felicity de Cogan, from the University of Birmingham’s Institute of Microbiology and Infection, have invented a method of delivering this otherwise-injected drug as eye drops.

Laboratory research showed that these eye drops have a similar therapeutic effect as the injected drug in rats. Now the scientists have taken their research one step further by investigating the effect of the eye drops in the larger eyes of rabbits and pigs, which are more similar to human eyes.

This latest study demonstrates that the eye drops can deliver a therapeutically effective amount of the drugs to the retina of the larger mammalian eye.

The technology behind the eye drops is a cell-penetrating peptide that can deliver the drug to the retina (the back of the eye). The scientists’ pending patents for the eye drops are now owned by US-based company, Macregen Inc, and a team of Birmingham researchers is working with the company to develop a novel range of therapies for AMD and other eye diseases.

The combined team is now expediting proof of concept studies to confirm the validity of the therapeutic approach. Clinical trials will be imminent once these studies are completed, and started as early as spring 2019.

2). „Nature Communications” 2020: Eye drops used to treat common cause of vision loss in mouse study.

Scientists at Columbia University have developed a potential new treatment in the form of eye drops that intervene in this process to preserve retinal function, with tests on human subjects now in the works.

 The Columbia University team has been working on potential solutions in mice, and recently made a very useful discovery that centers on the role of an enzyme called caspase-9.

This enzyme plays an important role in programmed cell death, a process whereby damaged or otherwise unnecessary cells are marked for destruction, so they can be cleared from the body for fresh, healthy cells to take their place. Through their experiments in mice, however, the researchers found that when retinal vein occlusion takes hold and damages blood vessels, the activity of caspase-9 spirals out of control and in turn injures the retina. So the team experimented with a new type of therapy, in which a highly selective caspase-9 inhibitor was worked into eye drops that were administered to the mice. This topical treatment had the effect of dampening caspase-9 activity and protecting the function of the retina, by reducing swelling, boosting blood flow and preventing damage to the all-important photoreceptor cells.

“We believe these eye drops may offer several advantages over existing therapies,” says Columbia University’s Carol M. Troy, who led the research. “Patients could administer the drug themselves and wouldn’t have to get a series of injections. Also, our eye drops target a different pathway of retinal injury and thus may help patients who do not respond to the current therapy.” Buoyed by these promising results in mice, the team has now set its sights on human subjects with preparations underway for phase 1 clinical trials. It is also hopeful that targeting caspase-9 in this way could bring about new treatments for other conditions caused by its over-stimulation, which include stroke and diabetic macular edema, another cause of blindness.

3). HORIZON 2020: Eye drops specially formulated to prevent and treat retinal damage

Diabetic retinopathy is a major cause of vision-loss globally. Spanish company Retinset has developed a novel drug product to tackle this problem, with eye drops that are now being readied for clinical trials thanks to backing from an EU project.

“With our RetinDR formulation, we are offering a therapeutic solution to these patients for the very first time. Available as eye drops, RetinDR is a topical ophthalmic formulation that effectively prevents the progression of the disease and regresses the symptoms of retinal neurodegeneration caused by diabetes,” says Marta Guerrero, from the EU-funded project RetinDR, and co-founder and CEO of Retinset.

Point 41.

545 – 547: I do not understand this sentence.

Answer 41.

Thank you for your remark. It is now:

Interlinking toxicity status involving glutamate and VEGF-induced blood-retina barrier breakdown is very important intriguing strands linking the phenomena of neuroretinal degeneration and vascular complications.

Reviewer 2 Report

The authors have addressed the comments that were made in my first review. They have included a comprehensive figure and this is appropriately referenced.

With the changes made in response to other reviewers the section outlining the structure and function of blood vessels appear out of place when the vascular complications of diabetes have already been discussed.  It would appear to be un-necessary at this stage in the review.

There are sections within the text where the font size is different.

Author Response

Response to Reviewer 2 Comments

Thank you for preparing the review of our manuscript ID: jcm-1035004. All your comments were taken into account.

Point 1.

With the changes made in response to other reviewers the section outlining the structure and function of blood vessels appear out of place when the vascular complications of diabetes have already been discussed.  It would appear to be un-necessary at this stage in the review.

Answer 1.

Thank you for your attention.

The section regarding the structure and function of blood vessels has been removed and was renamed” The neurovascular unit of the retina” to sort out the details refers to the structure and function blood vessels.

  1. The neurovascular unit of the retina

The retinal neurovascular unit includes the physical and biochemical relationship among neurons, glia, and blood vessels and the close interdependency of these tissues in the retina and the central nervous system. The neural unit (ganglion cells and glial cells) and the vascular unit (endothelial cells and pericyte) constitute the retinal neurovascular unit, together with retinal pigment epithelial (RPE) cells, they are the main body of BRB. The BRB is divided into two parts: the inner blood-retina barrier (iBRB) consists of retinal endothelial cells that are covered by astrocytes, pericytes, and Müller cells end-foot and is essential for maintaining the microenvironment of the inner layers of the retina; the outer blood-retina barrier (oBRB) is composed of tight junctions formed by neighboring RPE cells and serves as a filter to regulate solutes and nutrients from the blood. Hyperglycemia causes cells dysfunction, when impairment occurs, structural defect and functional disorder arise in the neurovascular unit, which in turn lead to further cells impairment [13, 62-64]. 

4.1. The neural unit

The neural unit includes retinal ganglion cells(RGCs), glial cells, and other neural cell types.

4.1.1. RGCs

The Integrity of RGCs provides the normal function of the retina, however, abnormal environment such as hyperglycemia alters the structure or function of RGCs and this kind of RGCs impairment is progressive in the development of subsequent DR.

4.1.2. Glial cells

There are three major types of glial cells in the retina and they are involved in maintaining retinal homeostasis: Müller cells, astrocytes, and microglia. 

Müller cells are the leading type of glial element, representing 90 % of the retinal glia. The inner limiting membrane is made up by the bases of Müller cells and connects to the base of the vitreous humor. The circular junctions between photoreceptors and Müller glial cells form the outer limiting membrane. Müller cells manage vascular responses to fulfill the metabolic demand of neurons, interchange metabolites, recycle neurotransmitters, and aid establishing the extracellular chemical environments [65]. It has been proved that the population of these cells shows traits of stem cells and is able to develop in different cells of the retina. The accelerated apoptosis of ganglion cells is accompanied by lesions within Müller glial cells. 

Hyperglycemia and inflammation induce activation of microglial cells that are found in the inner part of the retina and migrate later to the subretinal space and release cytokines, thus causing death of nerve cells. The activated microglial cells adhere to the vessels and seem to play a crucial role in vascular wall damage [66,67]. Similar results were obtained by Schellini et al., who investigated the morphology of Müller cells in the retina of healthy rats, with treated and untreated diabetes, 1 and 12 months following the induction of diabetes. In treated rats the lesions were less distinct [68-70]. It was also found that Müller cells in diabetic patients showed increased expression of glial acidic fibrilar protein (GFAP) [71]. Since Müller cells produce factors able to modulate blood flow, vascular permeability and cell survival, they are thought to have a major effect on endothelial function.

The connection between Müller cells and the vascular unit is affected during DR and they produce large amounts of factors, while some show protective effect, more are making it worse such as vascular endothelial growth factor(VEGF). VEGF is upregulated in the early stages of DR and presents a neuroprotective effect for the maintenance and survival of neurons, VEGF derived from Müller cells in diabetes can also increase blood circulation for proper metabolic activities [55-56].

Named based on their stellate shape, astrocytes are linked to the retinal blood vessels and neurons and play a significant role in the BRB integrity. In contrast to Müller cells, astrocytes are considered almost exclusively located in the innermost retinal layers. The distribution of astrocytes is related to the presence and position of retinal blood vessels, therefore, astrocytes are more likely to locate in the vascular regions of the retina, while avascular zones contain nearly no astrocytes [72]. Under the diabetic environment, characterized by dysmetabolic and hypoxic, astrocytes are activated and produce a variety of pro-inflammatory cytokines as well as chemokines, together with T-cells, monocytes/macrophages, and microglia, amplifying the inflammatory response [73]. 

As the resident inflammatory cells of the retina, in addition to immunologic activity, microglia also participate in neural network development and maintenance. In non-proliferative DR(NPDR), the increased perivascular microglia cells settled into the retinal plexiform layers and exhibit mildly hypertrophic. In PDR, microglial cells gathered in ischemic areas and new dilated blood vessels heavily. The contact between Microglia and RGCs are also found in epiretinal membranes. These findings suggest that activated microglia involved in all stages of DR and may even accelerate DR progression [74].

4.2. The vascular unit

Micro-vasculopathy is a typical characteristic of DM, which can eventually lead to vision loss. As the basic unit of retinal vascular structure and function, the stability of pericytes and endothelial cells is necessary for maintaining vascular integrity.

4.2.1. Endothelial cells (EC)

Endothelial cells (EC) (monolayer squamous epithelium) form the smooth internal vascular lining indispensable for normal blood flow. They play a major role in cardiovascular homeostasis maintenance, mainly through the regulation of arterial blood pressure. Apart from ensuring the barrier between the vessels and its lumen, endothelial cells secrete numerous mediators regulating platelet aggregation, fibrinolysis, coagulation and the tension of the vessels. Endothelial dysfunction observed in diabetic patients contributes to the development of lesions in large and small vessels. The major factors include hyperglycemia, insulin resistance, hyperinsulinemia and dyslipidemia. Endothelial cells secrete a few mediators that can alternatively mediate in vascular stenosis [75-77]. The complex tight junctions of ECs are one of the significant parts of the iBRB, which protects the retina from injuries by circulating substances or metabolites. Under the development of DR, increased capillary occlusion results in retinal ischemia and hypoxia, triggering an increase in the levels of VEGF, which then promotes pathological neovascularization. The receptors on ECs are activated by VEGF, and this kind of activation promotes ECs survival, proliferation, permeability, and taxis toward the VEGF sources, the essential steps of blood vessels production and maintenance, however, in many DR patients, VEGF is excessively produced in the retina resulting in the over-proliferation of ECs, new blood vessel formation, thus contributing to microaneurysms, fluid leakage, and tissue damage [78].

Endothelial progenitor cells (EPC) are immature cells able to differentiate into mature endothelail cells. Vascular damage and tissue ischemia mobilize the EPC by releasing growth factors and cytokines. The EPC stimulate compensatory angiogenesis chiefly at the sites of ischemia in the peripheral circulation. Furthermore, the EPC make up a circulating pool of cells capable of forming a “cellular patch” at sites of endothelial damage, simultaneously contributing directly to homeostasis and repair of the endothelial layer.  Tissue ischemia is regarded as the most powerful stimulus of EPC mobilization by means of activation of the systems that detect hypoxia, such as hypoxia-induced factor (HIF) -1α. Both patients with type 1 and 2 diabetes have fewer circulating EPC than healthy individuals. EPC in diabetic patients show the impairment of the function, such as decreased proliferation, adhesion, migration and a potential to be incorporated into vascular structures  [13, 76-77].

4.2.2. Pericytes 

Pericytes are the mural cells of blood microvessels, which have recently come into focus for modulating angiogenesis and maintaining BRB integrity [79]. Pericytes lying on the capillaries and are separated from epithelial cells by the basement membrane. They can prevent ischemia-reperfusion after thrombus clearance by constricting capillaries whereas their relaxation increases blood flow [80].  

Loss of pericytes, accompanied by ECs degeneration and the basement membrane thickening, may leads to the formation of acellular capillaries. These abnormal capillaries are inclined to generate microaneurysm and hemorrhage, and are related to vascular occlusion, or induced by thrombosis secondary to residual ECs impairment, resulting in ischaemic-hypoxic retinal damage [81]. 

Point 2.

There are sections within the text where the font size is different.

Answer 2.

Thank you for your remark. The font size was unified.

This manuscript is a resubmission of an earlier submission. The following is a list of the peer review reports and author responses from that submission.

Round 1

Reviewer 1 Report

Authors in the manuscript titled “Retinal vascular endothelial cell dysfunction and neuroretinal degeneration in diabetic patients” reviewed the pathogenesis of diabetic retinopathy, especially on retinal endothelial dysfunction and neuroretinal degeneration. The manuscript is not quite systematic, as its discussion on factors associated with retinal endothelial dysfunction and neuroretinal degeneration is inconsistent.

Major Critics:

- The authors aimed to review the pathogenesis of diabetic retinopathy, while the organization of the manuscript is not sufficiently legible. The pathogenesis of diabetic retinopathy is still unclarified, but there is a consensus that persistent hyperglycemia is initially involved in various mechanisms. Important pathogenesis like AGE or VEGF, which are also associated with retinal endothelial dysfunction, are not discussed. The flow if the manuscript is not systematic.

  • Advanced glycation end product (AGE) pathway is considered to be a major pathway involved in the breakdown of blood-retinal barrier in DR, but this is not at all discussed in this review.  
  • In the same context, ischemia is important pathogenesis but VEGF-associated pathway is not sufficiently discussed.

- I think the paragraph regarding fluorescein angiography is not necessary to this review regarding the pathogenesis of diabetic retinopathy, unless the authors want to describe clinical features of DR.

Minor Critics:

- Sentence not completed in line 23, page 1, Abstract.

- Multiple typo error associated with spaces: lots of double spaces throughout the manuscript.

- Introduction: Unification of the types of DM needed - mixture of type 1/2 and type I/II.

- For retinal structure: inner nuclear layer, not internal nuclear layer (line 70, page 2)

- Optic nerve is formed by retinal nerve fibers, not cells (line 79, page 2).

- Size description too definitive. For example, the optic disc diameter varies – please provide a range, rather than giving a number. Optic disc diameter is not always 1.6 mm. (line 81, page 2).

Reviewer 2 Report

This paper is a review of many aspects of diabetic retinopathy (DR).  It briefly touches on many ideas that have been put forward, and it has some value in bringing in aspects of cardiovascular function that eye researchers may not be familiar with.  However, it is largely uncritical in presenting these, and the paper lists a lot of ideas, with one or two references for each, when each of these fields is pretty large (for instance oxidative damage), and it does not attempt much of a synthesis.  There are better reviews of diabetic retinopathy already, and this one is unlikely to be much help to the field.  What might be useful would be a review of some particular smaller area – like endothelial cell dysfunction, and considering hypotheses for that alone. 

The paper contains very basic information about eye structure and function that are out of place here, and would be known already by anyone interested in diabetic retinopathy.  The figures, with the exception of one showing microaneurysms, are also not useful, but a schematic connecting some of the different threads would be useful.  The paper reads as though it was the background section for a PhD thesis, and it seemed like I was reading a draft that needed correcting.  In addition, there are many places where the English itself is incorrect, and fixing that is beyond what a journal copy editor could be expected to do. I do appreciate the difficulty of writing in English, but the authors probably need a native English speaker as a collaborator. Some examples are below.

Major:

1) it is frequently asserted that DR is an ischemic disease and in the later stages involving capillary loss, that is undoubtedly true. The increase in VEGF suggests this, and there is some evidence (the Amin et al reference) that VEGF increases in background retinopathy.  But, VEGF is controlled by many things in addition to HIF-1 that are not related to ischemia.  So it is not at all clear that ischemia is an early event, or, if it is, what would cause it.  Leukostasis is one possibility, but it has been very difficult to sort out what is important in disease progression, and what is a side effect.

2) l. 65-92: there is no point in a paper about DR in giving all this background about the retina.  The audience for this paper would be people who know something about the retina, and want to know the latest thinking about the cause or progression of DR. It may be useful to have the sections on EC and EPCs on the next page, but here too there are things that are too basic for such a paper, and there is no attempt to make this specific to the retina.  Researchers attempt to use retinal vascular endothelial cells in many cases, because they might differ from those elsewhere, although of course all of them have certain similarities.

3) I do not think anyone thinks aldose reductase plays much of a role in DR any more.  The two references for this are from 1997, and while Tilton et al did observe changes in sorbitol levels, there is something wrong here, because rats simply do not exhibit proliferative retinopathy.  I do not believe that they connected the sorbitol changes to structure or function, but could be wrong about that.

4) As the authors state, there is a big argument about whether vascular or neural lesions are first, and whether one is the cause of the other.  Section 12 refers to some papers on this topic, but does not give us any mechanisms to explain what the relationship would be except in very vague terms.

Minor (this is not an exhaustive list)

  • l. 9 and 33: diabetes is a societal problem, not a social problem
  • l. 14-18 this is basic vascular biology and is out of place here.
  • l. 18: the phrases in the sentence starting “endothelial cells…” do not have parallel structure
  • l. 23. Functional and structural are not connected to a sentence.
  • l. 37. While is not the right word
  • l. 53. “When the time of duration of diabetes” should just be “when the duration of diabetes
  • l. 109. ME, which must mean macular edema, is not defined.
  • l. 130. Angiogenesis already implies new growth. I think neoangiogenesis is a new and unnecessary word.  Sometimes neovascularization is used to mean new growth, which in the retina is always abnormal (unlike in some tissues).
  • l. 64. Ref 42 is about nephropathy, not retinopathy; the microvascular in different organs do not seem to be the same.
  • l. 208-09: There is one sentence about hyperoxia, but it isn’t clear what it is doing here. It is not connected to the ideas about NO.
  • l. 216: oxidative stress, not antioxidative stress.; at the end of this line, ROS- reactive oxygen species?
  • l. 242: microglia are immune system cells and are not the same as glial cells. I am not sure how this fits in.
  • l. 299: what neurohormonal mechanism are you referring to?
  • l. 350; 351; Glutamate is the main neurotransmitter; it does not belong to them. “Just elevated” under what conditions?
  • l. 372-373. The experiments with topical PEDF are interesting, but this was in mice. Mice have eyes that are only about 3 mm in diameter, and it is unlikely that topical PEDF would reach the retina in the human eye, where the posterior pole is 25 mm from the cornea. That distance is devastating for diffusion, even for a very small molecule like oxygen. And topical treatments are very likely to be cleared by the aqueous flow and uveal circulation. I do not know of any topical treatment for any eye disease that treats a retinal problem.
  • Ref 99 contains two references.

Reviewer 3 Report

Major comments:

The aticle reviews the mechanism(s) proposed to be central to the damage to the retina for individuals with diabetes.  It is ineteresting in that the authors seek to derive a link between the neural and the vascular mediated damage that results in diabetic retinopathy and pulls together seminal literature in this area.  The bulk of the review focuses on the vascular mediated changes and this reflects the origin of the bulk of the literature. 

However, it would be helpful to have an accompanying figure to illustrate the relationship the authors are proposing.  The authors should clarify the perspective of the pericytes as there are inconsistensies withn the text. Line 63 states that the pericytes are increasing in DR while in line 226 the authors state that pericytes do not replicate in the adult body  

Minor comments:

line 35: should read increase and not multiply.

line 37: should read with diabetes and not while diabetes.

line 77 - 78: should read protect rathan perform a protection.

line 73: should read two sources of vasculature rather than double vasculature.

line 126: should read cardiovascular rater than the cardiovascular.

line 165: spelling of retinopathy.

line 188: insert commas between CD34.....and EPC.

line 211: should read a signalling molecule.

line 215: should read an inbalance rather than occurs the inbalance and add occurs after processes.

line 350: jest elevated? clarify

line 352: should read activation rather than ctivation.

line 359: statement is incomplete.

line 386: should glutame read glutamate?

line 389: should read receptor rather than receptors and accelerated neurone death rather than the accelerated neurones death.